# An enhanced forest classification scheme for modeling vegetation-climate interactions based on national forest inventory data

Titta Majasalmi[1], Stephanie Eisner[1], Rasmus Astrup[1], Jonas Fridman[2], Ryan M. Bright[1]

[1]Norwegian Institute of Bioeconomy Research (NIBIO), 1431 Ås, Norway
[2]Swedish University of Agricultural Sciences (SLU), 901 83 Umeå, Sweden

*Correspondence to*: Titta Majasalmi (titta.majasalmi@nibio.no)

**Abstract.** Forest management affects the distribution of tree species and the age class of a forest, shaping its overall
structure and functioning, and in turn, surface-atmosphere exchanges of mass, energy, and momentum. In order to attribute climate effects to anthropogenic activities like forest management, good accounts of forest structure are necessary. Here, using Fennoscandia as a case study, we make use of Fennoscandic National Forest Inventory (NFI) data to systematically classify forest cover into groups of similar aboveground forest structure. An enhanced forest classification scheme and related Look-Up Table (LUT) of key forest structural attributes (i.e. maximum growing season leaf area index ($LAI_{max}$),
basal area weighted mean tree height, tree crown length and total stem volume) was developed, and the classification was applied for Multi-Source NFI (MS-NFI) maps from Norway, Sweden and Finland. To provide a complete surface representation, our product was integrated with the European Space Agency Climate Change Intiative's Land Cover (ESA CCI LC) map of present day land cover (v.2.0.7). Comparison of the ESA LC and our enhanced LC -products (https://doi.org//7zZEy5w3) showed that forest extent notably (kappa=0.55, accuracy=0.64) differed between the two
products. To demonstrate the potential of our enhanced LC-product to improve the description of maximum growing season LAI ($LAI_{max}$) of managed forests in Fennoscandia – we compared our $LAI_{max}$ map with reference $LAI_{max}$ maps created using the ESA LC-product (and related cross-walking table) and PFT-dependent $LAI_{max}$ values used in three leading land models. Comparison of the $LAI_{max}$ maps showed that our product provides spatially more realistic description of $LAI_{max}$ in managed Fennoscandian forests compared to reference maps. This study presents an approach to account for the transient nature of
forest structural attributes due to human intervention in different land models.

Keywords: Land use; land use change; forest management; LULCC; NFI; MS-NFI, PFT; SVAT; ESA CCI LC-product

# 1 Introduction

The structural properties of a forest largely determine the amount of mass, energy, and momentum exchanged with the atmosphere contributing to weather and climate at multiple scales (Bonan 2008). Given their controls on photosynthesis, albedo, and evapotranspiration, structural attributes like canopy leaf area and tree heights are crucial variables in modeling of carbon, water, and energy fluxes in forests. Leaf Area Index (LAI, defined as hemisurface area of foliage per unit horizontal ground surface area (Chen and Black, 1992)) is considered an Essential Climate Variable (ECV) (GCOS, 2012) as it quantifies the areal interface between the land surface and the atmosphere – hence representing a control over the exchange of mass and energy between the terrestrial biosphere and the atmosphere (Bonan, 2015). Similarly, canopy top and bottom heights $z_{top}$ and $z_{bottom}$ are important variables required for calculating roughness length and displacement height that largely determine aerodynamic resistances to heat, moisture, and momentum transfer (Oke, 2002). In land models, land surfaces are often classified by the main aggregate Land Cover (LC) classes: vegetation, urban, inland water, bare soil and ice – typically with the assistance of optical satellite remote sensing (Friedl et al., 2002; Hansen et al., 2000; Poulter et al., 2015). The 'vegetation' LC-class is further divided into a number of sub-classes according to their biophysical properties, grouping them into what is often termed Plant Functional Types (PFTs) – or "broad groupings of plant species that share similar characteristics (e.g. growth form) and roles (e.g. photosynthetic pathway) in ecosystem function" Wullschleger et al. (2014). LC-maps are converted into PFT maps using various model-dependent algorithms (e.g., Lawrence and Chase, 2007; Reick et al., 2013) or 'cross-walking' tables (e.g. ESA LC, 2017, manual p.75; Poulter et al., 2015).

Differences in forest structure within a given LC type (or PFT) can differ substantially (Kuuluvainen et al., 2012; Newton, 1997), and preserving within-LC (or within-PFT) differences in forest structure is necessary for more accurate modeling of surface fluxes in forests. While some land models assimilate local information on present day forest structure from satellite remote sensing to account for within-PFT variation (e.g., Community Land Model 4.5 (CLM4.5, Oleson et al., 2013) and Jena Scheme of Atmosphere Biosphere Coupling in Hamburg (JSBACH, Reick, 2012)), future structure must be still prescribed. Because land use transitions in modeling simulation studies of anthropogenic LC change are often represented by a change in PFT area in land models, post-disturbance changes to structure within forest PFTs go undetected (Lawrence et al., 2012; Reick et al., 2013). Hence, a forest classification that accounts for major variation in key structural attributes such as LAI or canopy height may lead to better predictions of surface fluxes in forests – not only in studies of prescribed land cover/management change, but also for dynamic vegetation studies that rely on fixed PFT parameters obtained from Look-Up Tables (LUTs). The time-invariant nature of the fixed parameter LUTs may be avoided by increasing the number of forest classes within a single forest PFT with sufficient differentiation in key structural attributes (i.e. from young to mature forests). In addition to grouping forests according to their shared phenological characteristics, further grouping according to their structural characteristics (i.e. accounting the effects of forest management) would strengthen prediction confidence in intensively managed regions.

LC data needed by land models should ideally be representative of sufficiently large areas, because incorporating fragmented data stemming from individual research sites and individual research experiments is limited by available computing resources and rapidly increasing model complexity. Nowadays most countries are conducting National Forest Inventories (NFIs) to quantify the extent and amount of forest resources with standardized reporting for compiling global Forest Resources Assessments (FRAs, e.g. FAO, 2015). Earlier, NFI data has been used in research aiming to attribute climate effects to management activities, because it reflects the human influence on forest structure (Bright et al., 2014; Naudts et al., 2015; 2016). However, as NFI data characterizes only forested areas, other LC data is needed to form a complete surface representation required by land models. The state-of-art LC-products, such as the European Space Agency (ESA) Climate Change Initiative's (CCI) LC -product (ESA LC, 2017), allow cross-walking from LC-classes to PFTs used in land models. Noteworthy is that ESA LC-products have high spatial resolution (0.003°) compared to common global grid sizes (e.g. 0.5°-1°) used in land models, which allows more flexibility in cross-walking and LC data aggregation.

In this study, we develop a forest classification scheme based on NFI data to better characterize the transient nature of forest key structural attributes to provide more realistic starting values for different land model simulations. We develop our concept using NFI data from Fennoscandia as it represents one of the most intensively managed forested regions of the world (e.g. Kuuluvainen et al., 2012). From the perspective of climate modeling, NFI data is well-suited for enhancing the structural description of forests in global LC datasets, because similar data is available for most developed countries and new data is collected systematically. The aims of this study are to: 1) develop a semi-objective clustering analysis approach to come up with a LC-dependent structural LUT, which reflects the transient nature of the key forest structural attributes in managed forests, 2) create a new forest LC-product for Fennoscandia based on Multi-Source NFI (MS-NFI) data to allow spatial application of the LUT of key structural attributes, 3) augment the new forest classification by importing non-forest LC-classes of the ESA LC-product to form a complete surface representation required by land models, 4) compare forest extent of the new LC-product with the original ESA LC-product to point out geographic areas where the largest differences occur to provoke discussion on alternative information sources for parameterizing land models, and 5) visualize and compare maps of our maximum growing season LAI ($LAI_{max}$) map with reference $LAI_{max}$ maps produced using the ESA LC-product, a model-generic cross-walking table (Poulter et al., 2015), and PFT-dependent $LAI_{max}$ values used in three land models: JSBACH, Joint UK Land Environment Simulator (JULES) and Organizing Carbon and Hydrology in Dynamic EcosystEms (ORCHIDEE).

**2 Materials and Methods**

**2.1 Data**

**2.1.1 NFI plot data**

Norwegian NFI data (Tomter et al., 2010) from 2007 to 2015 and Swedish NFI data (Fridman et al., 2014) from 2011 to

2015 were used in this study. NFI employs a network of field plots from which trees are measured and the growth is monitored systematically. NFI data are systematically collected and processed by forest authorities and are used to quantify the amount and extent of forest at national level. The diversity in forest structure throughout the Fennoscandian region is well-represented in the Swedish and Norwegian NFI data. The Norwegian NFI contained data from 10,813 circular 8.92 m radius sample plots (250 m$^2$), while the Swedish data contained data from 14,032 circular 10 m radius sample plots (314 m$^2$).

Plots which were divided (i.e. not completely circular) or which did not have trees were excluded from the data prior to the analysis. The main tree species of the area are Norway spruce (*Picea abies* (L.) H. Karst.), Scots pine (*Pinus sylvestris*, L.), and Silver and Downy birches (*Betula pendula* Roth and *pubescens* Ehrh.). Monocultural plots of birch are rare, but birches are common in plots with different species mixtures. Plot data were classified as spruce, pine or deciduous (contains also other tree species) dominated forests based on species with the largest share of total stem volume (m$^3$/ha) on the sample plot

(**Table 1.**).

**2.1.2 MS-NFI data**

NFI plot characteristics are extrapolated for areas between the NFI plots using non-parametric k-Nearest Neighbor (kNN) estimation method (e.g. Tomppo et al., 2014). The extrapolation step is called Multi-Source NFI (MS-NFI), because it employs data from different remote sensing systems (i.e. satellite and aerial platforms) and NFI plots. MS-NFI applies high

spatial resolution (<900 m$^2$) satellite images to separate forested areas from other LC-classes and digital terrain models to correct topographical distortions. In Fennoscandia, common stand size is only 1-2 ha (i.e. landscape is fragmented by forests with different development stages) and thus high spatial resolution satellite products are needed to prepare the MS-NFI maps. All processing of the MS-NFI data is done by forest authorities. MS-NFI maps are typically provided for forest variables such as Lorey's height (i.e. basal area weighted mean tree height) (H, m), forest stand age (years), and stem volume

(V, m$^3$/ha) by species. The newest MS-NFI maps for Finland (of 2013) and Sweden (of 2010) were downloaded from Natural Resources Institute Finland (LUKE) portal (LUKE, 2016) and from Swedish University of Agricultural Sciences (SLU) portal (SLU, 2016). For Norway, MS-NFI data (compiled during the first decade of the twenty-first century) called 'SAT-SKOG' (Gjertsen, 2007; 2009) and a forest resource map called 'AR5' (Ahlstrøm et al., 2014) were used to obtain all required inputs and coverage of the northernmost forest areas (i.e. Finnmark county) (**S1**).

### 2.1.3 ESA LC-product

The ESA LC-product series contains set of annual maps from 1992 to 2015 (ESA LC, 2017). The product series follows processing in which a baseline LC-product is applied; although the annual maps are not fully independent from each other, they are temporally consistent. The baseline LC-product was based on MERIS FR and RR archive between the years 2003-2012, and was back- and updated based on data from other satellite sensors (AVHRR between 1992-1999, SPOT-VGT between 1999-2013, and PROBA-V between 2013-2015). In this study, the newest 2015 (v.2.0.7) LC-product was used as it is more likely to be used by land modelers. This LC-product version has been validated against GlobCover 2009 reference data (cf. p. 39 in the product manual (ESA LC, 2017)). The spatial resolution of the ESA LC-product is ~0.003º, and it follows standardized hierarchical classification by United Nations Land Cover Classification System (UN-LCCS), which allows conversion between LC-classes into PFTs based on a cross-walking table (ESA LC, 2017, manual p.75; Poulter et al., 2015). The 2015 ESA LC-product contains three LC-classes to describe forests in Fennoscandia: broadleaved deciduous (60-62), needleleaved evergreen (70-72), and mixed broadleaved and needleleaved (90) (ESA LC-labels in parentheses). Second label digit from the left is designed to indicate forest fraction within a LC-pixel: The canopy is 'closed' when forest pixel cover fraction is >40% (labels 61 and 71) or 'open' when forest pixel cover fraction is between 15-40% (labels 62 and 72). Labels 60 and 70 are used to indicate that within pixel forest fraction is more than 15%, but it is not known whether that pixel is closed or open. The 2015 ESA LC-product clip for Fennoscandia contained only classes 60, 61, 70 and 90 (i.e. no pixels were assigned to subclasses 62, 71 or 72).

### 2.2 Methods

### 2.2.1 Forest classification scheme

NFI data were first used to develop the forest classification scheme based on four key forest structural attributes: total stem volume (V), Lorey's height (H), crown length (CL) and $LAI_{max}$ ($LAI_{max}$ calculation is described in **S2**) (**Fig. 1**): V defines species dominance, H corresponds with the aerodynamic height or $z_{top}$ (Nakai et al., 2010), CL is needed to estimate canopy bottom height or $z_{bottom}$ (i.e., $z_{bottom} = H - CL$; modeling of CL described in **S3**), and $LAI_{max}$ quantifies the exchange surface area between the land surface and the atmosphere. First, a clustering analysis of the NFI data was used to find medoids of the four-dimensional (4d): V-H-CL-$LAI_{max}$ -clusters, because forest variables are not independent from each other. After defining the 4d cluster centers, another Euclidean distance based classifier was used to define the 4d-cluster boundaries in order to apply the classification on MS-NFI data. Overview of the analysis is shown in **Fig. 1.**

The Norwegian and Swedish NFI data were merged and grouped into species groups (i.e. spruce, pine, and deciduous dominated) to account for differences in forest structural properties. First, the optimal number of structural subgroups within each species group (i.e. spruce, pine, deciduous) was analyzed. The optimal number of clusters was assessed by plotting the curve between total within-cluster sum of squares (wcss) and the number of clusters (k), and observing around which k the

relationship resembles a 'bend knee' (i.e. 'elbow method' (Ketchen and Shook, 1996)). Analysis was run using R package 'factoextra' (Kassambara and Mundt, 2017). In each species group the bent knee was located between k=3 and k=5, and thus the optimal k was set to 4 (i.e. the number of structural subgroups was set to four). Then, using the predefined number of structural subgroups (k=4) within each species group (n=3), a k-medoids clustering analysis was used to define the cluster

'centroids' (i.e. medians) within V-H-CL-$LAI_{max}$ -space to form a LUT of the key forest structural attributes (n × k =12 forest classes). The k-medoids algorithm belongs to the data partitioning methods in which each cluster is represented by one of the cluster objects (i.e. all subgroup LUT values (V, H, CL and $LAI_{max}$) are from the same plot) (Kaufman and Rousseeuw, 1990). The k-medoids algorithm assigns all plots to the nearest cluster centers and calculates the wcss. New cluster centers are updated and the plots are reassigned. Cluster centers are adjusted iteratively until they did not change. The

k-medoids algorithm was chosen because only the number of clusters is required as an input, and also because it is robust against outliers. The analysis was run using the 'cluster' package in R (Maechler, 2017).

A method to assess cluster boundaries was needed, because many plots were located near the edges of the 4d-clusters. We chose to determine cluster boundaries using V and H since these are often available for large geographical areas from MS-

NFIs. Mahalanobis (1963) Distance (MD) was used to quantify the within-cluster variation within V- and H–space (i.e. VH-space), because it corresponds to the Euclidean distance after V and H have been normalized. MD is a multidimensional method to determine how many standard deviations a data point is away from the class mean. For MD calculation the cluster mean values were obtained based on the 4d (i.e. V, H, CL and $LAI_{max}$) and not in 2d (i.e. V and H) clusters, and thus the cluster boundaries are not circular. MD values were calculated for each species group and the respective subgroups. The

binning (i.e. grid of 14 × 14) interval of the VH-space was set subjectively to add resolution on younger forest structures. For each grid cell, and for each subgroup, a median MD value was calculated. To represent results using a grid surface, the cell was assigned to a subgroup with the smallest median MD.

### 2.2.2 Compiling the enhanced LC-product

In order to apply the newly developed LUT, the most recent MS-NFI maps from Norway, Sweden and Finland were used to

classify the Fennoscandian forests into the twelve forest classes (i.e., the forest LC-product was developed). MS-NFI data were classified as spruce, pine or deciduous dominated based on species with the largest share of pixel total stem volume ($m^3$/ha). The share of other tree species than pine or spruce was assigned to deciduous group. After the species group was assigned, a gridded VH-space was used to determine pixel subgroup. Possible VH-combinations without MD value (i.e. falling outside the VH-space) were assigned to the closest subgroup based on V. After classifying all data, the forest classes

were recoded as integers between one and twelve (i.e. three species groups × four structural subgroups).

The classified MS-NFI maps were reprojected, aggregated, and resampled to complement the ESA LC-product (v.2.0.7) (ESA LC, 2017) to form a complete surface representation required by land models. Two types of aggregation routines were

used for upscaling: forest class was assigned based on mode (among the twelve forest classes), and within-pixel forest cover fraction based on mean (for this purpose, forested pixels in MS-NFI data were recoded as 100 and other pixels as 0). In other words, for each forested ESA LC-pixel ($\sim$90 000 m$^2$), forest class and within-pixel forest cover fraction (%) were obtained based on classified MS-NFI maps ($\sim$260 m$^2$ or $\sim$630 m$^2$). Two exceptions occurred: 1) If the ESA LC-pixel was not
classified as forest, but the MS-NFI maps indicated the presence of forest, and 2) If the ESA LC-pixel was classified as a forest but MS-NFI maps indicated non-forest. For pixels which were classified as forest by the ESA LC-product, but forest cover fraction within that pixel in the enhanced LC-product did not exceed the 15% threshold (i.e. definition used by the ESA LC-product) according to the MS-NFI data, forest class was assigned based on moving average interpolation (referred as 'gapfilling'). Gapfilling was necessary because land (climate) models require completeness in LC to resolve computations
of mass, energy, and momentum fluxes (Note, gapfilled pixels are coded separately which allows them to be either included or excluded from analysis). Non-forest LC-classes were imported from the ESA LC-product to supplement our forest pixels.

To allow for more flexible cross-walking or aggregation to a lower spatial resolution in the preparation of surface datasets in land models, for each forest LC-pixel, coverage fractions (referred as 'percentage layers') for each of the twelve forest
classes were calculated (i.e. twelve layers with values ranging between 0 and 100 based on subgroup abundance within the ESA LC-pixel). In addition, gapfilled pixels and non-forest LC-classes are provided as own layers. These layers also allow more flexibility by modelers in choosing the number of desired input land cover classes (i.e., modelers may use e.g. three most abundant forest classes instead of keeping to the most abundant one). Raster analyses were performed using 'rgdal' (Bivand et al., 2017) and 'raster' (Hijmans, 2017) packages in R.

**2.2.3 Comparison of the LC-products**

To highlight areas where the forest extent differed the most, a difference map between the enhanced 'back-classified' (i.e. into ESA LC-classes) map and ESA LC-product was calculated using $\sim$0.3° resolution. This resolution was chosen as it represents a good compromise between higher resolution regional modeling (0.05° – 0.1°) and coarser resolution regional/global modeling (0.5° - 1°). In addition, LC-class changes (e.g. from cropland to conifer forest) were quantified
using a confusion matrix between the ESA LC-classes and enhanced back-classified map classes in the original product resolution ($\sim$0.003°). The back-classification was done using the percentage layers of different forest subgroups: If >=70% of the MS-NFI pixels within the ESA LC-pixel were classified into conifer or deciduous group, the pixel was classified as 'needleleaved' (class 70) or 'broadleaved' (class 60), but otherwise it was classified as 'mixed' (class 90). The difference map was calculated after both products (i.e. the enhanced back-classified map and the ESA LC-product) were aggregated to
$\sim$0.3° resolution using pixel modes. The LC-class changes s are presented with a confusion matrix between the ESA LC-product and the enhanced back-classified map; Each row of the matrix represents an occurrence in the enhanced back-classified map, while each column represents the respective occurrence in the ESA LC-product. Percentage of pixels belonging to the same LC-class in both products is shown in diagonal, whereas percentage values outside the diagonal

quantify the change ('confusion') between the different LC-classes. The confusion matrix was calculated using 'caret' package (Kuhn, 2008) in R.

### 2.2.4 Comparison of $LAI_{max}$ maps

To demonstrate the potential of our LC-product to provide more realistic LC-dependent values of key forest structural attributes for forested areas in Fennoscandia, we compared our $LAI_{max}$ map (i.e. produced using enhanced LC-product and related LUT) with reference $LAI_{max}$ maps produced using the ESA LC-product, a model-generic cross-walking table (Poulter et al., 2015), and PFT-dependent $LAI_{max}$ values used in ORCHIDEE, JULES and JSBACH. Only forested pixels were used for demonstration.

According to ESA LC cross-walking table (Poulter et al. 2015), a pixel classified as Broadleaved Deciduous Trees (BDT) is assumed to contain 70% of BDT, 15% of Broadleaf Deciduous Shrub (BDS) and 15% of natural grass. A pixel with Needleleaved Evergreen Trees (NET) is defined to comprise 70% of NET, 5% of Broadleaf Evergreen Shrubs (BES), 5% of Broadleaf Deciduous Shrubs (BDS), 5% of Needleleaf Evergreen Shrubs (NES), and 15% of natural grass. As all the required shrub LUT values were not available, the percentage of BDS was set to 15% for NET (i.e. equivalent to BDT weights). For a mixed leaf type forest (ESA LC-class 90), the estimates were obtained as an average of BDT and NET values, because all the required input data (e.g. values for Needleleaf Deciduous Tree (NDT) and BES) were not available to follow the ESA LC-product cross-walking scheme.

JULES, JSBACH and ORCHIDEE LUTs employ PFT-dependent $LAI_{max}$ values. In JULES the PFT-dependent $LAI_{max}$ values are: 9 for a Broadleaf tree (BDT), 5 for a Needleleaf Tree (NET), 4 for a C3 grass (natural grass) and 3 for a Shrub (BDS) (acronyms used by the ESA LC cross-walking table in parenthesis) (Clack et al., 2011). In ORCHIDEE the respective PFT-dependent $LAI_{max}$ values are 4.5 for both boreal BDT and NET, and 2.5 for a C3 grass (same $LAI_{max}$ also used for BDS) (Lathiere et al., 2006). In JSBACH the PFT-dependent $LAI_{max}$ value was 5 for extra tropical deciduous tree (BDT), 1.7 for coniferous evergreen trees (NET) and 3 for C3 grass (same $LAI_{max}$ also used for BDS) (Schürmann et al., 2016).

## 3 Results

### 3.1 Forest classification scheme

As a result of our classification scheme, a LUT of the key structural variables (i.e. V, H, CL, and $LAI_{max}$) was created (**Table 2.**). The boundaries of the subgroups were determined based on MD, which can be visualized using a gridded representation of vegetation subgroups within the VH-space (**Fig. 2**) to select the right values from the LUT. The classified grid area and subgroup membership patterns reflect the variability of V and H in NFI data, which was used to define the classes. For example, the spruce dominated plots may have V up to 1500 $m^3$/ha. In pine dominated plots the V did not exceed 900 $m^3$/ha, and in deciduous plots the highest V was 1100 $m^3$/ha. In spruce dominated plots, the H exceeded 30 m with many different Vs, whereas for pine the 30 m was exceeded either when the respective V was less than 50 $m^3$/ha (i.e. tree is left for seed production during harvesting which is a common forest regeneration strategy in Fennoscandia) or large (more than 500 $m^3$/ha). In plots dominated by deciduous species the 30 m is exceeded after V was more than 150 $m^3$/ha. The location and size (i.e. patterns) of different subgroups in VH-space cannot be directly compared between different species groups, as Euclidean distances were used for their classification.

### 3.2 Enhanced LC-product

The majority (58%) of the forest pixels in Fennoscandia were classified as pine dominated which was also the largest species group in Sweden (58%) and in Finland (72%) (**Table 3.**). However, in Norway the largest species group was deciduous broadleaf (42%). Finland had slightly higher percentage of deciduous forests than Sweden. Spruce dominated forest was the smallest species group in Finland (14%). Visual assessment of spatial distribution of different species groups and their subgroups showed that low-land areas in Finland and in Sweden were mainly dominated by pines and spruces, whereas deciduous species were most abundant in the northernmost, mountainous and coastal areas (**Fig. 3**). In Fennoscandia the most abundant subgroup within spruce dominated forest was 'Spruce 3' (i.e. species group = 'spruce' and subgroup number = '3') with class median values of V= 201 $m^3$/ha and H= 17 m (see **Table 2.**). Within the pine dominated forest the most abundant subgroup was 'Pine 2' with class median V= 80 $m^3$/ha and H= 12 m. For deciduous species group the median values of the largest subgroup 'Deciduous 1' were V= 7 $m^3$/ha and H= 5 m.

### 3.3 Forest extent comparison

In order to assess agreement between the two LC-products, the enhanced LC-product was back-classified into ESA LC-classes using the percentage layers of different forest subgroups. Kappa coefficient (measure of agreement which takes into account possible agreement occurring by chance) for classification was 0.55, and classification accuracy was 0.64 (calculated based on **Table 4.**). The confusion matrix between the ESA LC-product and the enhanced back-classified LC-map showed that the highest agreement (30.3%) between the two classifications schemes occurred for forest class 70 (i.e.

NET) (**Table 4.**). The largest fraction of forested pixels in the enhanced back-classified LC-map were classified as conifer dominated (=36.3%) (**Fig. 4**). Results showed that 1.5% of class 70 was classified into class 60 (i.e. BDT), and 14.4% of ESA LC-class 70 was classified as 90 (i.e. 'mixed' of BDT and NET). The share of mixed forest class (i.e. class 90) was also high (=30.2%). However, the portion of pixels classified as BDT was relatively low (=5.5%). Overall, the enhanced LC-product contained 16.4% more forest classified pixels than the ESA LC-product (Note, the fraction of gapfilled forest pixels was 4.5%). Forest area increased by 4.8% at the expense of ESA LC-class shrub or herbaceous cover (class 180), 4.1% at the expense of LC-class mosaic tree and shrub (>50%) (class 100), 2.3% at the expense of LC-class croplands (class 10) and 1.9% at the expense of LC-class water bodies (class 210). The classified land area increased by 0.5% as areas classified as no-data in the ESA LC-product were classified as forest in the enhanced LC-product.

The spatial distribution of different LC-classes and the class frequencies of the enhanced back-classified map are shown in **Fig. 4**, which shows both ESA LC-class labels and descriptions (Note, ESA LC-product class frequencies are also shown for reference). The difference map of forest cover between the enhanced back-classified LC-product and the ESA LC-product pointed out that the areal representation of forests differ the most in mountainous areas in Norway and Sweden, south and north Finland, and in mid-south Sweden (in area between Stockholm and lakes Vättern and Vänern (lakes visible in see **Fig. 4.**) (**Fig. 5.**). The areas classified as forest by the ESA LC-product, but not by MS-NFI data ('Gapfilled forest' in **Fig. 5.**), were mainly located in mountainous areas in Norway and Sweden.

## 3.4 LAI$_{max}$ map comparison

The enhanced LC-product and related LUT produced a Fennoscandic mean LAI$_{max}$ of 3.2. Spatial variations in LAI$_{max}$ appeared more natural when compared to the reference maps (**Fig. 6**). The standard deviation (sd) – a measure of the variability -- of LAI$_{max}$ was substantially larger (sd=2.3) for the enhanced LC-product that for the three reference LAI$_{max}$ maps. The smallest mean LAI$_{max}$ values (mean LAI$_{max}$= 2.4, sd=0.7) were produced by LAI$_{max}$ values applied in JSBACH. The LAI$_{max}$ map produced using ORCHIDEE had a mean of 3.9 (sd=0.0 due to applied cross-walking scheme and equal LAI$_{max}$ values for NET and BDT). The largest mean LAI$_{max}$ was produced by LAI$_{max}$ values applied in JULES (mean LAI$_{max}$= 4.9, sd=0.9). Noteworthy is that the use of JSBACH and JULES PFT-dependent LAI$_{max}$ values produced unnaturally high LAI$_{max}$ values for the northernmost areas dominated by deciduous species.

## 4 Discussion

This paper is a response to the 'call to action' raised in the review by Ellison et al. (2017) which highlighted an urgent need to integrate forest effects on energy balances, hydrology, and climate into policy actions regarding climate change adaptation and mitigation. One of goals of this paper is to foster interdisciplinary discussions on alternative information sources, such as the existing NFI data, to enhance representation of forest structures in different land modeling frameworks. Although NFIs from different countries have been shaped by the local information needs, the work done by the Food and Agriculture Organization (FAO) in conducting global FRAs since 1948 has aided in developing national forest reporting standards. Currently, new assessments are carried out every five years and the 2015 assessment covered 93.5% of the global forest area (Köhl et al., 2015). Thus, as the aim of the FRA is to describe the state and change of the world's forests and keep policy makers informed, the same data could potentially be used to describe the current state of forests in land models. In this study, we developed a simple clustering and classification scheme to allow reiteration of our approach to NFI data from other countries. Classifying forests based on the structural properties they share at various successional stages under similar management conditions may be one way to link models of forestry with the land models employed in climate research. Important transient effects could then be included, for example, through changes in area under a given successional stage, with forestry models providing the link to the time dimension. Alternatively, distinct rule sets for successional dynamics following management disturbances could be developed analogous to those which are used to govern growth and competition in dynamic vegetation models (or land models run in dynamic vegetation mode).

Recently, other approaches have been developed for incorporating forest management into existing land surface (climate) models. For example, the radiative transfer based land-surface model ORCHIDEE was parameterized to simulate the effects of forest management for biogeochemical and biophysical variables (Naudts et al., 2015). The model was parameterized using diameter-at-breast-height (dbh) data from different European NFIs (French, Spanish, Swedish and German) (i.e. the key input values were modeled based on dbh using allometric models), and twelve parameter sets for specific tree species (instead of presenting groups of species such as PFTs) were presented. However, a major drawback of individual tree-based approaches is that existing global LC-products are not designated to distinguish between individual species, which limits the spatial domain where such approaches can be applied. In addition, the need for residual groups remains because individual tree based approaches are not suited for areas where the forests are essentially mixtures of different tree species. The benefit of defining 'broader' PFT classes, such as those developed in this study, is that the broad functional types may be separated from optical satellite data based on differences in optical and structural characteristics of the forests. In the future, as both the spectral, spatial, and temporal resolution of the optical satellite data improves, definition of narrower forest classes may be justified. Alternatively, Functional Traits (FT) may be used for modeling vegetation-climate interactions (Wullschleger et al., 2014; Verheijen et al., 2013). Commonly, the community-weighted-mean trait value (i.e. based on relative abundances of species and their trait values) is used in models which apply the FTs concept. While FTs are highly scalable (i.e. from

organism to ecosystem scale), well assembled (i.e. leaf, stem and root traits), and measurable (at least in theory), the downside of FTs is that their applicability is in its infancy, and the lack of standards hinders its practical application. In addition, many traits such as root traits cannot be measured using remote sensing.

At present, some countries, such as Finland and Sweden, have national Airborne Laser Scanning (ALS) campaigns producing high resolution forest structural data which could be used to obtain more accurate forest height estimates or to develop forest classification schemes for different land models. However, the drawback of these ALS datasets is that they cannot be used to separate between different tree species, which is one of the most important forest structural attributes. In addition, as few countries have national ALS datasets, the geographical extent which could be covered using ALS based
forest classification schemes would remain limited. At present, the use of optical satellite data to classify the forests is unquestionable due to its superior spatial and temporal resolution, and thus will probably sustain its role as the most valuable tool for environmental monitoring and mapping. While Synthetic Aperture Radar (SAR) allows more robust and temporally continuous data collection compared to optical instruments (i.e. SAR is not limited to cloudless conditions unlike optical instruments) – the relatively low spatial resolution ($km^2$) cannot be used to separate different aged forests in landscapes
which are fragmented (into 0.01 $km^2$ units) by active forest management. Data from SAR could be used to harmonize MS-NFI data from different countries, and to provide other land model inputs such as soil moisture maps. In the future, approaches combining both optical and ALS/SAR data may be expected to become more common, and thus allow development of more sophisticated forest classification schemes to increase the accuracy of the climate predictions.

The forest extent differs significantly between the enhanced LC-product and the ESA LC-product because they employ different forest definitions. ESA LC-product is based on series of satellite surface reflectance data (i.e. between years 1992-2015) and LC-class is deduced based on pixel reflectance properties. However, processing of the MS-NFI data employs a forest mask which delineates potential forest areas prior the kNN-estimation (i.e. clear-cuts and harvest are seen as a natural part of forest development, and thus pixel inside forest mask may have V=0 or H=0) (**S4**). For example, it is not clear if a
sapling stand with e.g. V=3 ($m^3$/ha) and H=3 (m) would classify as forest based only on its reflectance. Thus, the enhanced LC-product cannot be directly used to validate the ESA LC-product. In addition, differences in forest extent are propagated by different spatial resolution of the input reflectance data (i.e. probability of having 'mixed' class pixels is higher using lower resolution data) and data aggregation using the mode (i.e. the most abundant classes will become more common). The influence of spatial resolution of the input data and the applied data aggregation method may be observed, for example,
around water bodies in Finland and in Sweden (e.g. **Fig. 4.** and **5.**). For example, a single ESA LC-pixel (~90 000 $m^2$) classified as water (located next to a larger water body), may contain more pixels classified as forest than water in high resolution (<900 $m^2$) MS-NFI data (e.g. Huang et al., 2002), and thus be classified as forest if data is aggregated using mode. Forest area of the enhanced LC-product is also larger than in the ESA LC-product, because MS-NFI data was complemented

with the ESA LC-product data, and pixels which were classified as forest by the ESA LC-product, but did not contain forest according to the MS-NFI data, were gapfilled.

The presented forest classification scheme has many levels to serve the needs of different users (**S5**). For example, for climate and hydrological modeling requiring full spatial coverage, the gapfilled pixels and non-forest LC-classes are provided. Researchers that are able to run their models with no data may select to remove the gapfilled pixels prior to analysis. Remote sensing scientists may wish to use only 'true' forest pixels and extract areas belonging to different species groups or subgroups, or select areas where the fraction of forests is lower or higher (i.e. 'open' or 'closed' following ESA LC-product legend definitions). In addition, the percentage layers – or the relative abundance of different forest subgroups within each LC-pixel in MS-NFI data -- provides land modelers more control and flexibility in terms of the number of input LC-classes in different land models: The percentage layers for different forest subgroups may be used to obtain complete LC-distributions for Fennoscandia, or alternatively, a modeler may choose, for example, to use three of the most abundant forest classes instead of holding onto the most abundant forest class. The percentage layers also provide more flexibility for cross-walking (Poulter et al., 2015) across different spatial resolutions. Our forest classification scheme and the related map products (i.e. enhanced LC-product and the percentage layers) allow customized model 'inputs' to fit the needs (or requirements) of various land models.

Earlier, development of cross-walking tables have been complicated by the fact that number and definition of PFTs used in today's land models varies, and the limited availability of spatially and temporally representative input LC datasets (Poulter et al., 2015). The new ESA LC-product series was the first to overcome the aforementioned challenges and demonstrate how to compile a LC-product series which optimally serves the needs of various users. Our work in this paper does not imply that the ESA LC-product and cross-walking scheme (or the model- and PFT-dependent parameter values) are necessarily wrong *per se*, but simply presents an alternative approach to classification that explicitly accounts for the structural variation of managed forests in different successional stages. The flexibility in spatial resampling of the ESA LC-product is preserved by our approach (i.e. the idea of our 'percentage layers' correspond with the 'fractional area' contribution employed by ESA LC-product) (Poulter et al., 2015), thus respecting the efforts of ESA LC-product developers. Our LUT-demonstration showed that by using the enhanced LC-product with its related LUT, the description of $LAI_{max}$ appeared more natural compared to $LAI_{max}$ maps of JULES, ORCHIDEE and JSBACH compiled using the ESA LC-product classification, PFT cross-walking table (Poulter et al., 2015), and model- and PFT-dependent $LAI_{max}$ values. While the accuracy of our product cannot generally be determined -- it presents a new approach to quantify the present state of the key forest structural attributes of managed forests in Fennoscandia. In regional modeling studies, based on our results, it appears worth the effort to use the enhanced LC-product instead of the original ESA LC-product when cross-walking from LC-classes to PFTs to obtain more truthful initial values of the key structural variables (i.e., $LAI_{max}$, $z_{bottom}$, $z_{top}$). The enhanced LC-product may be used for forecasting and back-casting the impacts of forest management on energy, water and carbon cycling; whether our

enhanced forest classification leads to improved regional climate predictions linked to transient changes occurring in forests over time remains the subject of future research activity.

To our knowledge, this is the first study to use NFI data together with MS-NFI maps to enhance the characterization of forest structure in a format that is compatible with many land surface (climate) models (i.e. in modeling frameworks) where changes in vegetation structure are captured by area-based changes in LC (or PFT). The methods used for creating the LUT were carefully explained to allow other researchers to replicate the same procedures using NFI data from other countries. The benefit of the classification scheme described in this study is that the required data (i.e. NFI data and MS-NFI maps of species, V and H) are readily available for many countries. Future research is needed to develop recommendations and guidelines for prescribing future forest transitions under changing climate and management regimes in different land models and modeling frameworks.

## Data availability

The MS-NFI Forest resource maps for Finland are available through Natural Resources Institute Finland (LUKE) portal: http://kartta.luke.fi/opendata/valinta.html. For Sweden the forest maps may be obtained through Swedish University of Agricultural Sciences (SLU) portal: http://www.slu.se/en/Collaborative-Centres-and-Projects/the-swedish-national-forest-inventory/forest-statistics/slu-forest-map/. For Norway the MS-NFI data are available by request from Norwegian Institute of Bioeconomy Research (NIBIO).

The enhanced LC-product for Fennoscandia, including the percentage layers, can be downloaded from: https://doi.org/10.21350/7zZEy5w3

## Acknowledgements

The research was funded by the Research Council of Norway, grant number 250113/F20.
We acknowledge the work done by ESA CCI Land Cover project, in addition to constructive feedback from three anonymous reviewers.

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

**Figures and tables**

**Table 1.** Descriptive statistics for the National Forest Inventory (NFI) data. Abbreviations: n=number of sample plots, dbh= diameter-at-breast-height, H=basal area weighted mean tree height (i.e. Lorey's height) and V=total stem volume.

| Area | Species | n | dbh (cm) mean | dbh (cm) range | H (m) mean | H (m) range | V (m³/ha) mean | V (m³/ha) range |
|---|---|---|---|---|---|---|---|---|
| Norway | Spruce | 3364 | 12.6 | 5.0 - 49.0 | 13.0 | 2.9 - 32.3 | 152.0 | 0.2 - 1492.4 |
| | Pine | 3650 | 14.1 | 5.1 - 48.9 | 11.5 | 2.4 - 28.6 | 97.8 | 0.2 - 656.6 |
| | Deciduous | 3799 | 9.4 | 5.0 - 99.9 | 8.3 | 2.4 - 24.8 | 53.0 | 0.2 - 592.9 |
| Sweden | Spruce | 4552 | 16.2 | 1.0 - 52.0 | 15.3 | 1.4 - 40.2 | 177.8 | 0.5 - 1010.2 |
| | Pine | 7028 | 16.2 | 1.0 - 64.6 | 13.6 | 1.4 - 32.1 | 120.0 | 0.6 - 752.3 |
| | Deciduous | 2452 | 12.8 | 1.0 - 81.2 | 12.8 | 1.5 - 32.6 | 101.9 | 0.4 - 1001.5 |

**Table 2.** A forest classification scheme Look-Up Table (LUT). Abbreviations: V=total stem volume (m³/ha), H=Lorey's height (m), CL=Crown length (m), and $LAI_{max}$=maximum growing season Leaf Area Index (m²/m²). Recoded label -column
10  is a key to be used with the enhanced CL-product. Interquartile range (i.e. first quartile subtracted from the third quartile) is given inside parentheses.

| Species group | Subgroup | Recoded label | V | H | CL | $LAI_{max}$ |
|---|---|---|---|---|---|---|
| Spruce | 1 | 301 | 22 (28.9) | 7.5 (3.1) | 6.3 (2.8) | 1.4 (1.6) |
| | 2 | 302 | 92.2 (51.7) | 12.3 (2.5) | 10.1 (2.2) | 4.3 (2.2) |
| | 3 | 303 | 201.3 (70.1) | 16.8 (3.1) | 13.2 (2.6) | 6.7 (2.5) |
| | 4 | 304 | 373.9 (138.9) | 22 (4.5) | 15.8 (3.5) | 9.1 (3.4) |
| Pine | 1 | 305 | 20.8 (23.1) | 7.5 (2.8) | 4.6 (1.7) | 0.9 (1) |
| | 2 | 306 | 80 (49.2) | 11.6 (2.4) | 6.7 (1.4) | 2.4 (1.4) |
| | 3 | 307 | 129.5 (67.9) | 17 (3.9) | 9.4 (2) | 2.3 (1.2) |
| | 4 | 308 | 236.4 (107.1) | 17.2 (5) | 8.4 (1.6) | 4.4 (1.5) |
| Deciduous | 1 | 309 | 7.2 (10.8) | 4.9 (1.6) | 3.2 (1.1) | 0.5 (0.7) |
| | 2 | 310 | 36.1 (28.9) | 8.4 (2.1) | 5.5 (1.3) | 1.8 (1.6) |
| | 3 | 311 | 97.6 (50.8) | 12.2 (3.7) | 7.9 (2.5) | 3.9 (2.1) |
| | 4 | 312 | 227 (111.2) | 18.3 (5.5) | 10.3 (3.2) | 7 (3.2) |

**Table 3.** The percentage (%) of forest pixels (i.e. excluding gapfilled pixels) belonging to different species groups in the enhanced LC-product (referred as 'Fennoscandia') and separately for each country (The spatial distribution of different forest subgroups and their frequency distributions are shown in **Fig. 3**). Values are based on MS-NFI data.

|           | Fennoscandia(%) | Norway (%) | Sweden (%) | Finland (%) |
|-----------|-----------------|------------|------------|-------------|
| Spruce    | 22.9            | 28.9       | 29.0       | 13.7        |
| Pine      | 58.1            | 28.9       | 58.3       | 71.7        |
| Deciduous | 19.0            | 42.3       | 12.6       | 14.6        |

**Table 4.** Confusion matrix in percentage (%) between the ESA LC-product and the enhanced back-classified LC-product. Grey background is used to indicate the ten classes with the highest percentage of pixels. Small percentages are shown as 0.00. Kappa coefficient was 0.55. Label definitions shown in **Fig. 4**. In order to compare the two LC-products the enhanced LC-product was back-classified into ESA LC-classes. Column names are the ESA LC-class labels and column sums represent the fraction of that LC-class present in the ESA LC-product. The row sums represent the fraction of that LC-class present in the enhanced back-classified map. Percentage of pixels belonging to the same LC-class in both products is shown in diagonal (e.g. the fraction of pixels classified into LC-class 70 is 30.3%). Percentage values outside the diagonal describe the change ('confusion') between different LC-classes (e.g. 14.4% of pixels classified into LC-class 70 by the ESA LC-product, were classified into a LC-class 90 in the enhanced back-classified map.

**ESA 2015 LC-product (v.2.0.7)**

Enhanced back-classified Land Cover Classification

|      | 0   | 10  | 11  | 30  | 40  | 60  | 70   | 80  | 90  | 100 | 110 | 120 | 122 | 130 | 140 | 150 | 152 | 160 | 180 | 190 | 200 | 201 | 210 | 220 | Sum: |
|------|-----|-----|-----|-----|-----|-----|------|-----|-----|-----|-----|-----|-----|-----|-----|-----|-----|-----|-----|-----|-----|-----|-----|-----|------|
| 0    |     |     |     |     |     |     |      |     |     |     |     |     |     |     |     |     |     |     |     |     |     |     |     |     |      |
| 10   |     | 2.5 |     |     |     |     |      |     |     |     |     |     |     |     |     |     |     |     |     |     |     |     |     |     | 2.5  |
| 11   |     |     | 0.3 |     |     |     |      |     |     |     |     |     |     |     |     |     |     |     |     |     |     |     |     |     | 0.3  |
| 30   |     |     |     | 0.2 |     |     |      |     |     |     |     |     |     |     |     |     |     |     |     |     |     |     |     |     | 0.2  |
| 40   |     |     |     |     | 0.1 |     |      |     |     |     |     |     |     |     |     |     |     |     |     |     |     |     |     |     | 0.1  |
| 60   | 0.0 | 0.0 | 0.0 | 0.0 | 0.1 | 2.8 | 1.5  |     | 0.3 | 0.3 | 0.1 | 0.0 | 0.0 | 0.0 |     | 0.1 | 0.0 |     | 0.3 | 0.0 | 0.0 |     | 0.0 |     | 5.5  |
| 70   | 0.1 | 0.2 | 0.0 | 0.0 | 0.1 | 0.7 | 30.3 |     | 1.6 | 1.4 | 0.1 | 0.0 | 0.0 | 0.0 |     | 0.1 | 0.0 | 0.0 | 1.4 | 0.0 | 0.0 |     | 0.3 |     | 36.3 |
| 80   |     |     |     |     |     |     |      | 0.0 |     |     |     |     |     |     |     |     |     |     |     |     |     |     |     |     |      |
| 90   | 0.4 | 2.1 | 0.1 | 0.4 | 0.4 | 1.8 | 14.4 | 0.0 | 2.2 | 2.5 | 0.5 | 0.0 | 0.1 | 0.0 |     | 0.3 | 0.1 | 0.0 | 3.1 | 0.2 | 0.0 | 0.0 | 1.6 |     | 30.2 |
| 100  |     |     |     |     |     |     |      |     |     | 1.3 |     |     |     |     |     |     |     |     |     |     |     |     |     |     | 1.3  |
| 110  |     |     |     |     |     |     |      |     |     |     | 4.8 |     |     |     |     |     |     |     |     |     |     |     |     |     | 4.8  |
| 120  |     |     |     |     |     |     |      |     |     |     |     | 0.0 |     |     |     |     |     |     |     |     |     |     |     |     |      |
| 122  |     |     |     |     |     |     |      |     |     |     |     |     | 0.3 |     |     |     |     |     |     |     |     |     |     |     | 0.3  |
| 130  |     |     |     |     |     |     |      |     |     |     |     |     |     | 0.1 |     |     |     |     |     |     |     |     |     |     | 0.1  |
| 140  |     |     |     |     |     |     |      |     |     |     |     |     |     |     | 0.0 |     |     |     |     |     |     |     |     |     |      |
| 150  |     |     |     |     |     |     |      |     |     |     |     |     |     |     |     | 7.0 |     |     |     |     |     |     |     |     | 7.0  |
| 152  |     |     |     |     |     |     |      |     |     |     |     |     |     |     |     |     | 0.1 |     |     |     |     |     |     |     |      |
| 160  |     |     |     |     |     |     |      |     |     |     |     |     |     |     |     |     |     | 0.0 |     |     |     |     |     |     |      |
| 180  |     |     |     |     |     |     |      |     |     |     |     |     |     |     |     |     |     |     | 3.0 |     |     |     |     |     | 3.0  |
| 190  |     |     |     |     |     |     |      |     |     |     |     |     |     |     |     |     |     |     |     | 0.3 |     |     |     |     | 0.3  |
| 200  |     |     |     |     |     |     |      |     |     |     |     |     |     |     |     |     |     |     |     |     | 1.9 |     |     |     | 1.9  |
| 201  |     |     |     |     |     |     |      |     |     |     |     |     |     |     |     |     |     |     |     |     |     | 0.1 |     |     |      |
| 210  |     |     |     |     |     |     |      |     |     |     |     |     |     |     |     |     |     |     |     |     |     |     | 5.8 |     | 5.8  |
| 220  |     |     |     |     |     |     |      |     |     |     |     |     |     |     |     |     |     |     |     |     |     |     |     | 0.3 | 0.3  |
| Sum: | 0.5 | 4.8 | 0.5 | 0.7 | 0.7 | 5.3 | 46.2 |     | 4.1 | 5.4 | 5.5 |     | 0.4 | 0.2 |     | 7.4 | 0.2 |     | 7.8 | 0.5 | 1.9 |     | 7.6 | 0.3 |      |

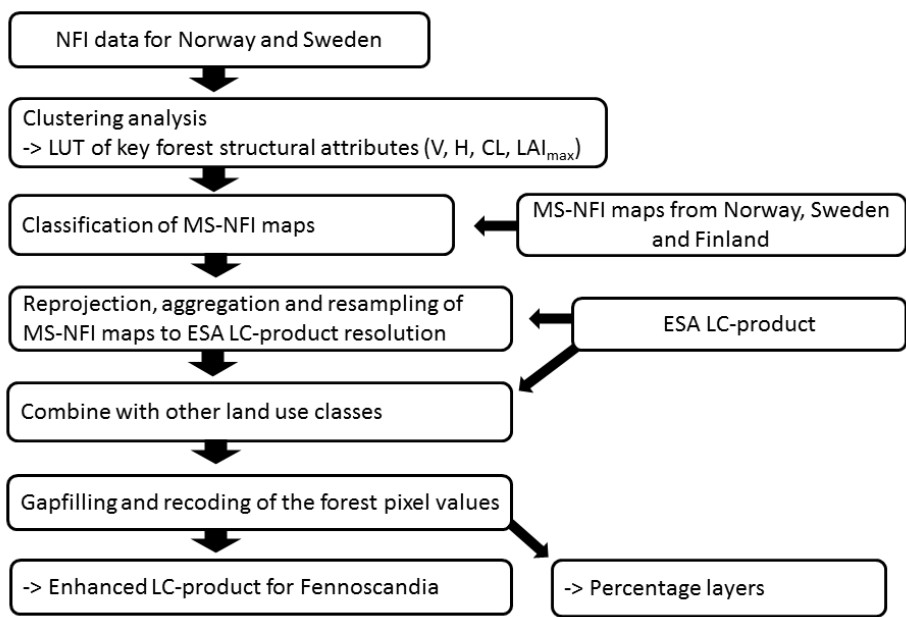

**Figure 1.** Flowchart for developing and applying the forest classification scheme (sections **2.2.1** and **2.2.2**). Abbreviations: National Forest Inventory (NFI), Look-Up Table (LUT), total stem volume ($m^3$/ha) (V), Lorey's height (H), Crown Length (CL), maximum growing season Leaf Area Index ($LAI_{max}$), Multi-Source NFI (MS-NFI, i.e. products provided by forest authorities), European Space Agency's Land Cover (ESA LC) product.

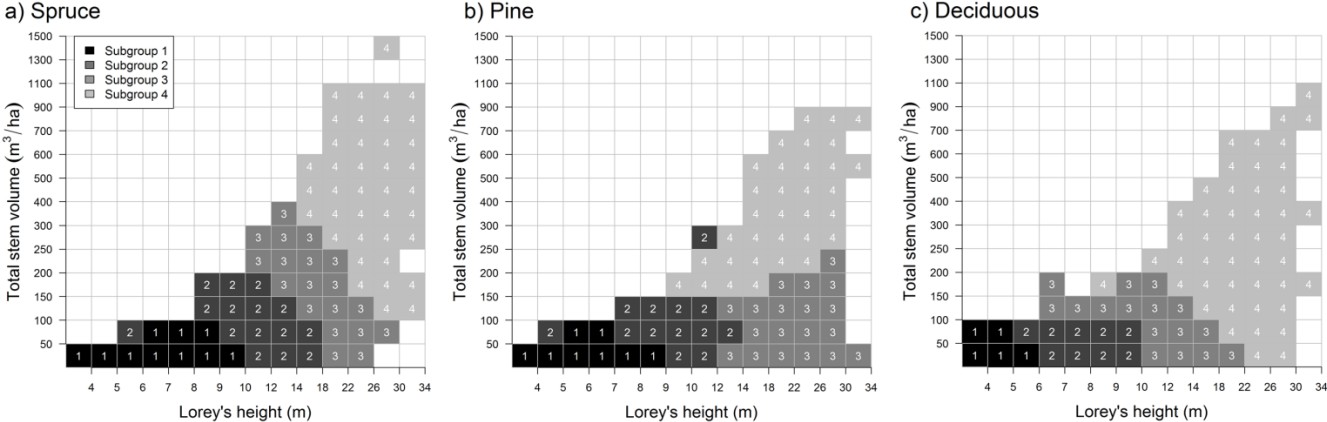

**Figure 2.** Gridded representation of vegetation subgroups, i.e. **a)** spruce, **b)** pine, and **c)** deciduous, within the total stem volume (V) and Lorey's height (H) –space (referred as VH-space) based on NFI data. Visualization is required to map subgroup distribution in VH -space and used to apply the classification to the MS-NFI maps.

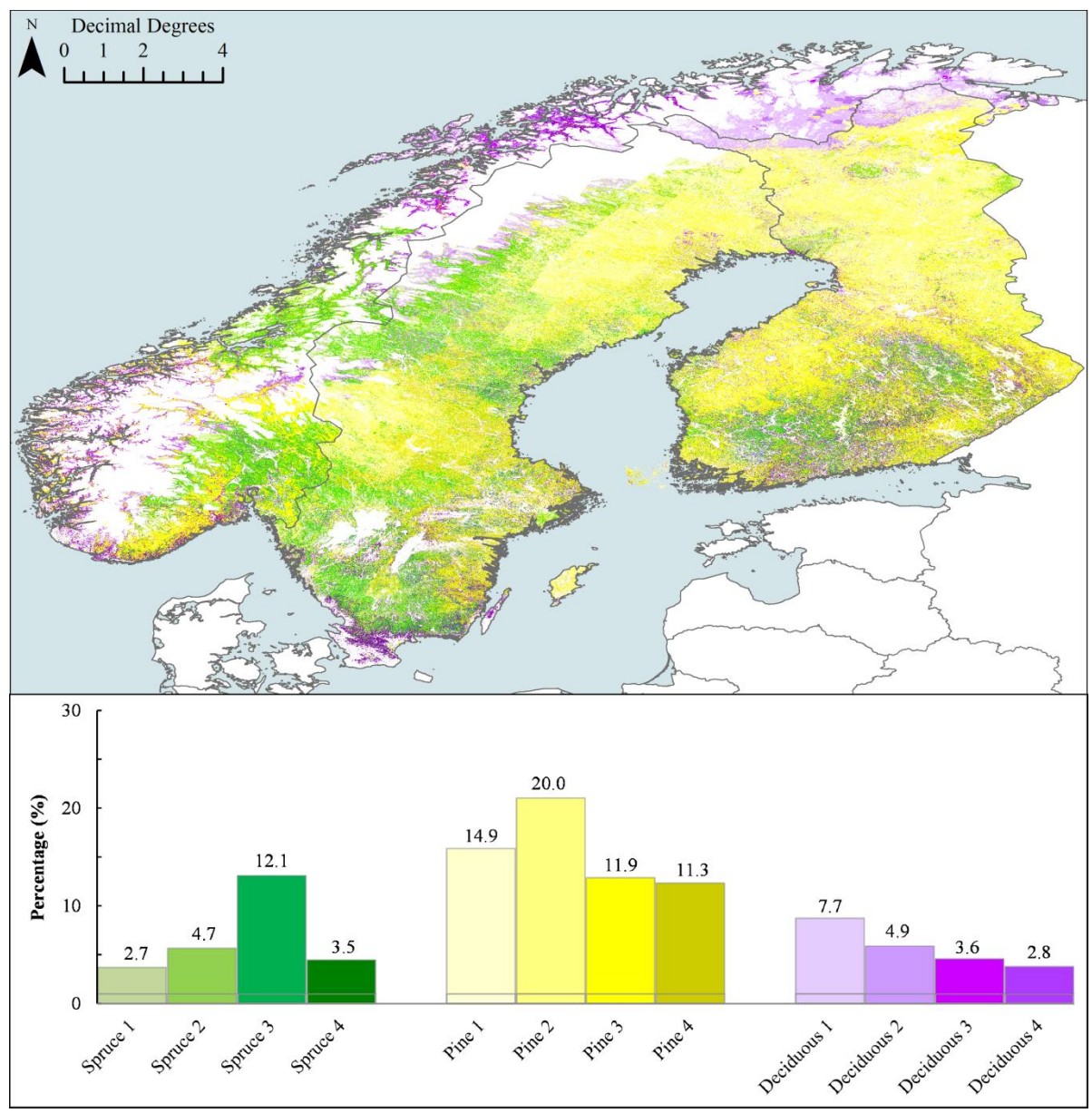

**Figure 3.** Spatial distribution of MS-NFI forest classes (i.e. without gapfilled forest pixels) in Fennoscandia. First part of the x-label is the species group and the number refers to the respective subgroup number (see **Fig. 2.**). The forest subgroup was assigned based on most abundant forest class within the ESA LC-pixel. For colors, see online version of the article.

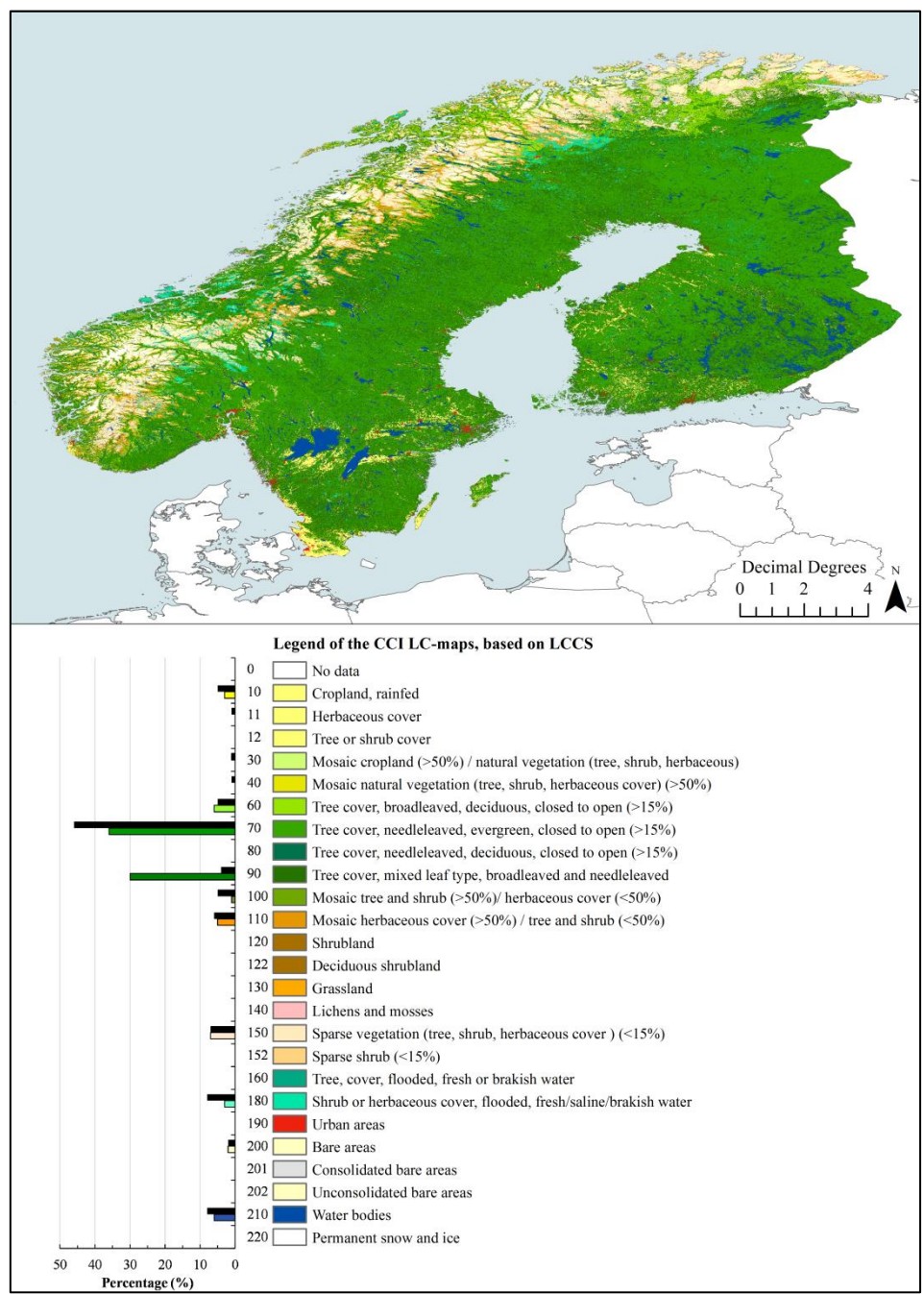

**Figure 4.** Enhanced back-classified map for Fennoscandia. The percentage layers of forest subgroups were used to back-classify the data into ESA LC-product classes (see section **2.2.2.**). Histograms show LC-class percentages for the enhanced back-classified map (lower bars with colors) and for ESA LC-product (black upper bars). For colors, see online version of the article.

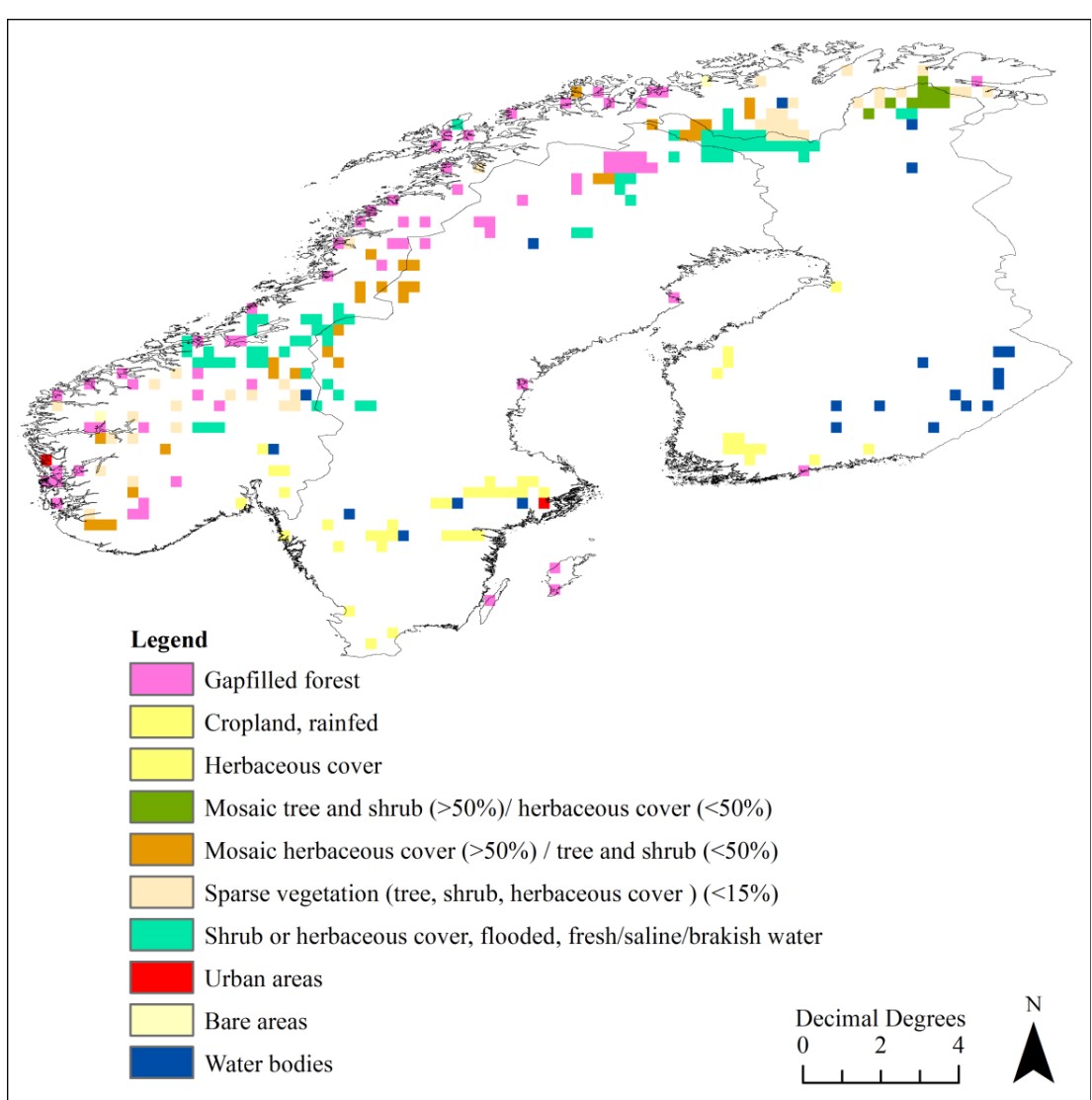

**Figure 5.** Difference in forest extent between the enhanced back-classified map and the ESA LC-product. Both data were aggregated to ~0.3° resolution using mode to display the main differences in the two classifications spatially. Label 'gapfilled forest' is used to indicate areas which were mainly classified as forest by the ESA LC-product, but the MS-NFI data indicated non-forest. Other labels (see **Fig. 4.**) show which main ESA LC-classes were classified as forest in the enhanced back-classified map. Note, the confusion matrix between the ESA LC-product and enhanced back-classified map was prepared using ~0.003° resolution (**Table 4.**), whereas the map resolution shown here is ~0.3°). For colors, see online version of the article.

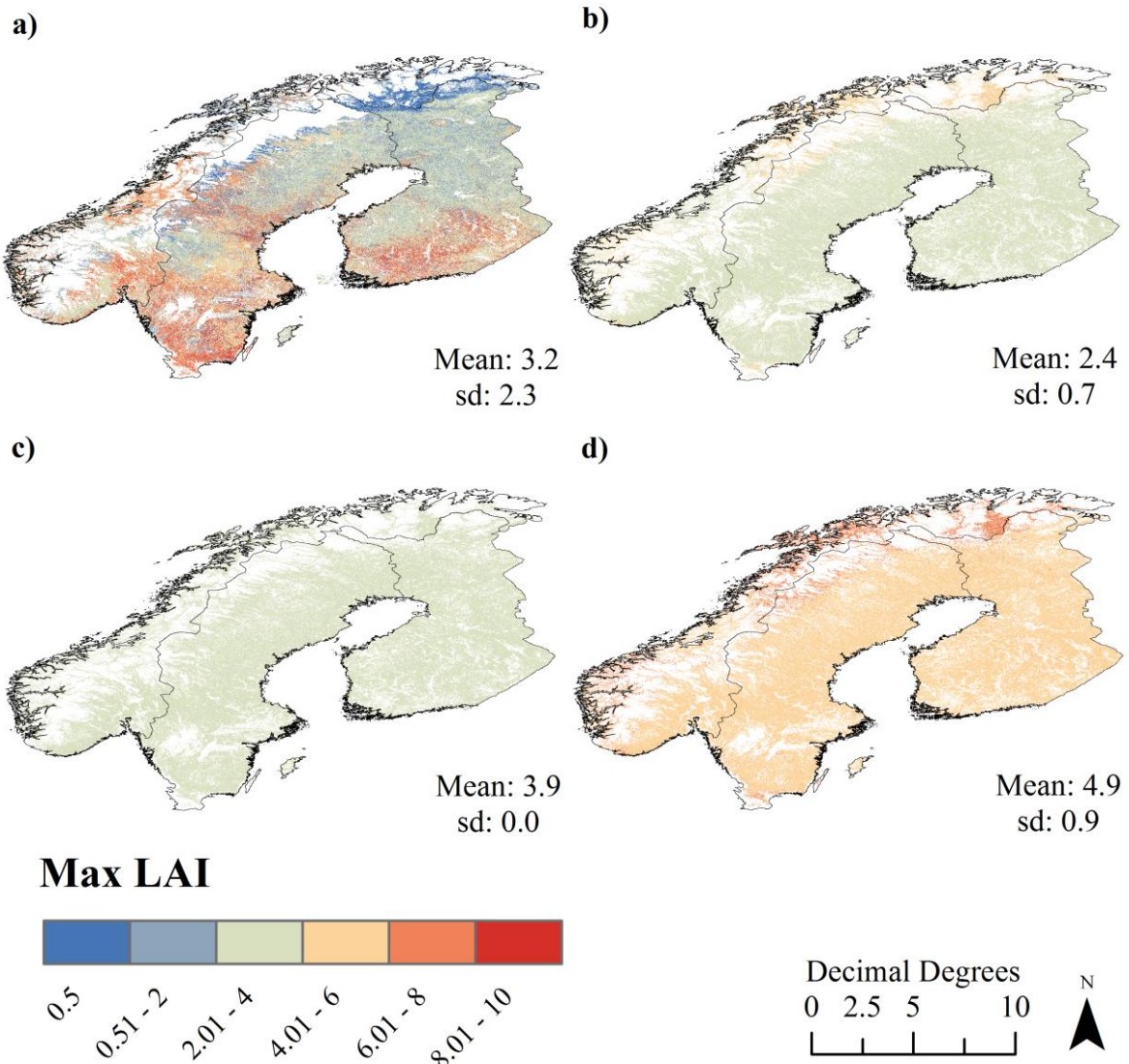

**Figure 6.** Demonstration on how the enhanced LC-product and the related LUT may be used to map local variations in important structural attributes in forests, such as maximum growing season LAI (max LAI or $LAI_{max}$). Maps of $LAI_{max}$ in Fennoscandic forests using: **a)** our enhanced LC-product and the related LUT, **b)** ESA LC-product and PFT-dependent $LAI_{max}$ values used in JSBACH, **c)** ESA LC-product and PFT-dependent $LAI_{max}$ values used in ORCHIDEE, and **d)** ESA LC-product and PFT-dependent $LAI_{max}$ values used in JULES. For colors, see online version of the article.

**Supporting information**

**S1. Preprosessing of Norwegian MS-NFI data**

As the northernmost forest area (i.e. Finnmark county) is not covered by SAT-SKOG (Gjertsen, 2009) a forest resource map called 'AR5' (Ahlstrøm et al., 2014) and NFI field plot data were used to model the forest cover information for Finnmark. The AR5 forest mask was used to define the forest extent in Finnmark. A moving window interpolation was used to fill in data gaps between the NFI field plots located inside the forest maks. As SAT-SKOG does not contain tree height information, it was modeled based on available data of tree species, tree age at breast-height (i.e. 1.3 m above ground

surface), and site index using equations by Tveite and Braastad (1981). Separate equations were used for pine, spruce, and birch (birch model was applied to all deciduous species). To calibrate the mean height to correspond with the Lorey's height (H), a separate model was developed based on Norwegian NFI data to scale the plot mean height into H. The H model was developed based on Norwegian NFI field plot data to predict the H based on plot median tree height ($H_m$) and plot total stem volume (V) (**Table S1.**). Median tree was used instead of mean tree while developing the model, because median tree exists

in the data and is less affected by outliers than the mean. Coefficient of determination (variance) explained by fixed effects was 0.82, and both fixed and random effects was 0.98. Model RMSE was 1.40. The model was not applied if either pixel $H_m$ or V was zero.

**Table S1.** Description of the Lorey's height model ($H \sim H_m + V + \varepsilon$.). Abbreviations: H=Lorey's height (m), $H_m$=median tree

height (m), V=plot total stem volume ($m^3$/ha), sd=standard error and $\varepsilon$ =residual.

| Area | Species | | Fixed effects: value | Random effects: sd (plot) | $\varepsilon$ |
|------|---------|---|------|------|------|
| Norway | All | Intercept | 3.015 | 1.841 | 0.692 |
| | | $H_m$ | 0.756 | | |
| | | V | 0.013 | | |

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

The highest mean canopy LAI values (LAI$_{canopy}$) were found from spruce dominated plots and the smallest in plots dominated by either pine or deciduous species (**Table S2.**). For conifer dominated plots the mean LAI$_{canopy}$ values were higher in Norway than in Sweden. The influence of small trees (LAI$_u$) on plot mean LAI were systematically larger in Norway than in Sweden, due to differences in definition of small trees in NFI measurements (i.e. in Norwegian NFI trees with dbh less than five centimeters are recoded as small trees, whereas in Sweden the breast height (1.3 m) threshold is used to count small trees). In addition, as Norway has alpine birches growing in mountainous areas the influence of LAI$_u$ on plot total mean LAI was larger in Norway than in Sweden.

Our results showed that the LAI$_u$ accounted for on average 2-8 % of total LAI, and thus should be taken into account while estimating the estimating the forest LAI. The contribution of LAI$_u$ on total LAI was the largest in deciduous plots, which are

in the center of climate change mitigation and adaption by forest management. To our knowledge, this is the first paper to report how the information of small trees in the NFI data may be used to estimate LAI of understory trees to approximate forest total canopy LAI. Error propagation of LAI estimates is not possible, because the 'true' LAI is not known (for details see Majasalmi et. al., 2013). Note, in our paper the sum of $LAI_{canopy}$ and $LAI_u$ is called $LAI_{max}$, and it may be interpreted as

the maximum growing season LAI of all forest canopy layers.

**Table. S2.** The mean Leaf Area Index (LAI) of plots dominated by different species groups: $LAI_{canopy}$ is the forest canopy LAI, $LAI_u$ refers to LAI of small trees i.e. understory trees (Note, the different counting method of small trees in Norway and in Sweden), and $LAI_{u\%}$ is the percentage of $LAI_u$ for the total mean LAI ($LAI_{canopy}+LAI_u$).

| | Norway | | | Sweden | | |
|---|---|---|---|---|---|---|
| | $LAI_{canopy}$ | $LAI_u$ | $LAI_{u\%}$ | $LAI_{canopy}$ | $LAI_u$ | $LAI_{u\%}$ |
| Spruce | 5.59 | 0.24 | 4.05 | 4.90 | 0.10 | 1.91 |
| Pine | 2.58 | 0.12 | 4.34 | 2.31 | 0.07 | 2.83 |
| Deciduous | 2.33 | 0.21 | 8.31 | 2.76 | 0.15 | 5.17 |

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

**S3: Crown ratio models**

Crown Length (CL) can be modeled directly or alternatively via Crown Ratio (CR), which is defined as CL divided by h (i.e. CL = h*CR). We used a mixed effects modeling approach with maximum likelihood estimator to develop the CR models for the three main species (i.e. spruce, pine, and birch) in Norway and in Sweden. Separate CR models were developed for Norway and Sweden to account for possible differences in forest management (i.e. harvesting) and environmental conditions. The number of trees used to develop the CR models was nearly 24,000 (i.e. spruce: 10,403, pine: 7,830 and deciduous: 5,828) in Norwegian NFI data. In Swedish NFI data the number of trees used to compile the models was almost 29,500 (i.e. spruce: 12,228, pine: 13,171 and deciduous: 4,081). The model was created based on forest variables which are influenced by forest management operations and are widely available from forest inventory databases (i.e. species, h, and V). The number of predictor variables was selected based on the Akaike Information Criterion (AIC), which quantifies the trade-off between model complexity and fit. To account for non-independencies between observations (i.e. several model trees from one plot) model intercept was allowed to vary between individual trees and plots. CR models were developed based on NFI trees for which both h and CL were measured (**Fig. S1**).

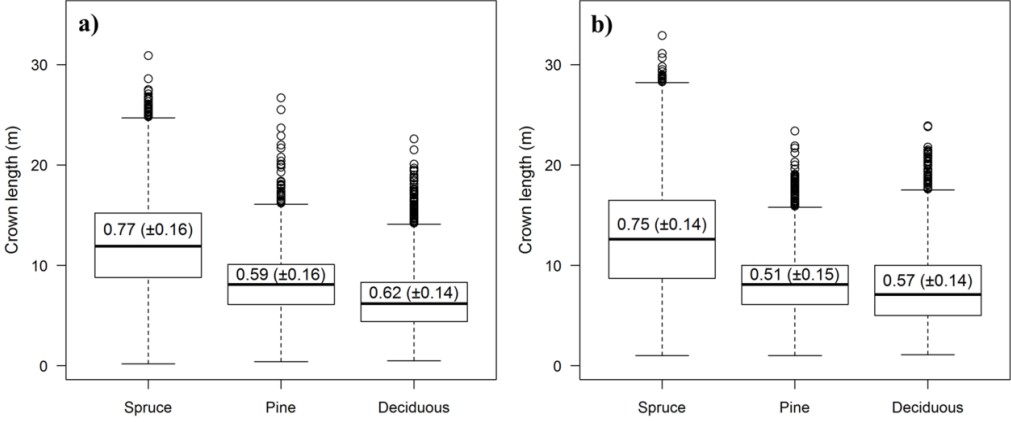

**Figure S1.** Individual tree data for developing the Crown Ratio (CR) models based on **a)** Norwegian National Forest Inventory (NFI) trees, and **b)** Swedish NFI trees. Mean and standard deviation (±sd) of measured CR are plotted inside the boxes. Box contains 50% of the values and dark horizontal line denotes the median value.

The fixed effect of the CR was modeled based on h and V (**Table S3.**). Both h and V were found to be significant predictors for CR (p=0.00) for each species (i.e spruce, pine, and birch). The variance explained by fixed effects of the CR models varied between 0.11-0.37 (**Table S4.**). In Norway, the largest portion of variance explained by fixed effects was noted for spruce CR model, whereas in Sweden the pine CR model explained the highest portion of variance of fixed effects. For both countries the variance explained by fixed effect remained the lowest for birch CR models. Including both fixed and random effects increased the portion of explained variance to range between 0.45 and 0.63.

The relatively low explanatory power of the CR models is due to large scatter of CR estimates. Higher prediction power could have been obtained by developing models for CL, which has larger dynamic range compared CR. However, by modeling CR instead of CL, the problem of negative CLs was avoided, and as a tradeoff we had to accept the lower prediction power of our CR models. CL reflects the vigor of trees and is influenced by both stand density via competition over time and environmental factors (e.g. water availability and temperature). Increasing stand density reduces CL due to decreasing light conditions near the crown base (i.e. foliage respiration exceeds the net photosynthesis) and mechanical damage as tree canopies collide with each other. Thus, thinnings have important role in controlling CL development and hence to the aerodynamic properties of forests.

**Table S3.** Description of the Crown Ratio (CR) models ($CR \sim h + V + \varepsilon$.). Abbreviations:  h=tree height (dm),V=plot total stem volume (m$^3$/ha), sd=standard error and $\varepsilon$=residual.

| Area | Species | | Fixed effects: value | Random effects: sd (plot) | sd (tree) | $\varepsilon$ |
|---|---|---|---|---|---|---|
| Norway | Spruce | Intercept | 0.821 | 0.055 | 0.055 | 0.100 |
| | | h | 0.008 | | | |
| | | V | -0.001 | | | |
| | Pine | Intercept | 0.659 | 0.067 | 0.067 | 0.102 |
| | | h | 0.005 | | | |
| | | V | -0.001 | | | |
| | Birch | Intercept | 0.612 | 0.058 | 0.058 | 0.105 |
| | | h | 0.008 | | | |
| | | V | -0.001 | | | |
| Sweden | Spruce | Intercept | 0.802 | 0.058 | 0.056 | 0.101 |
| | | h | 0.006 | | | |
| | | V | -0.001 | | | |
| | Pine | Intercept | 0.618 | 0.065 | 0.066 | 0.091 |
| | | h | 0.001 | | | |
| | | V | -0.001 | | | |
| | Birch | Intercept | 0.606 | 0.059 | 0.059 | 0.105 |
| | | h | 0.005 | | | |
| | | V | -0.001 | | | |

**Table S4.** Variance explained by fixed effects and both fixed and random effects of the Crown Ratio (CR) models. RMSE is the Root Mean Square Error.

| Area | Species | RMSE | Fixed effects | Fixed + Random effects |
|------|---------|------|---------------|------------------------|
| Norway | Spruce | 0.11 | 0.37 | 0.61 |
| | Pine | 0.11 | 0.26 | 0.60 |
| | Birch | 0.11 | 0.11 | 0.45 |
| Sweden | Spruce | 0.11 | 0.20 | 0.51 |
| | Pine | 0.10 | 0.24 | 0.63 |
| | Birch | 0.10 | 0.14 | 0.47 |

## S4. Background information concerning MS-NFI data

The forest masks used to apply the kNN are developed by forest authorities, and thus mask definitions may slightly vary between countries. In addition, statistical calibration is applied at municipality level to correct errors in the map data. The calibration is based on a confusion matrix between land use classes of the sample plots recorded in the field and extracted from the raster map. The forest area on the NFI maps is not corrected, and thus it is commonly overrepresented on the map (i.e. a pixel inside forest mask may have V=0 or H=0. Note, however, that in our classification such pixel would not be classified as forest). Due to local calibration, the values from the NFI maps do not directly correspond with the official forestry statistics. If forest authorities would provide these calibrated MS-NFI maps instead of the uncalibrated maps the MS-NFI estimates and forest statistics would agree. The underlying assumptions of our classification scheme are that NFI data represents different forest types within a country (i.e. samples the whole population) and that MS-NFI data represent the extent and structural variation of forests spatially. MS-NFI data are intended to represent large forest areas, and while the errors at pixel level are relatively high at fine spatial resolution, the error decreases as the size of the estimation area increases. For example, in Finnish MS-NFI the average error of the V estimates at pixel level is 57.8 m$^3$/ha and H is 4.6 m (errors calculated as an average of mineral soil and peatland estimates for Finnish MS-NFI 2009 products and are reported in MS-NFI 2013 metadata). According to Tomppo et al. (2014): "For a sufficiently large area consisting of a group of pixels, e.g., for areas of 200 000–300 000 ha, the MS-NFI estimates are compared to the estimates and error estimates based solely on field data". As the land area of e.g. Finland is 33,842,400 ha, and majority of the land area is covered by forest, the estimation errors of the MS-NFI data may be assumed small.

Tomppo, E., Katila, M., Mäkisara, K. and Peräsaari, J. The Multi-source National Forest Inventory of Finland - methods and results 2011. Metlan työraportteja / Working Papers of the Finnish Forest Research Institute 319. 224 p. http://www.metla.fi/julkaisut/workingpapers/2014/mwp319.htm, 2014.

## S5. Enhanced LC-product and percentage layers

In the final product, the forest pixels are recoded as follows: The twelve forest classes were recoded by adding 300 (e.g. "Deciduous forest subgroup 4" would be coded as 312 - see **Table 2.**). In addition, two digits were added after recoding the forest class number to indicate the presence of forest cover within the LC-pixel and whether the pixel is a 'true' forest pixel based on MS-NFI data data or gapfilled. The fourth digit is used to indicate the fraction of forested pixels within an LC-pixel: Value '1' indicates that the fraction of forested pixels within an LC-pixel is >40%, and value '2' denotes that the fraction of forested pixels within an LC-pixel is between 15-40%. For 'true' forested pixels the last digit is '0', whereas for gapfilled pixels the last digit is '1'. For example, a pixel with value "30210" can be interpreted as "Spruce, subgroup 2, >40% forest cover fraction, "true-forest" (i.e. non-gapfilled).

For land models which allow more than one land cover type, the percentages of each subgroup for each LC-pixel are provided as separate layers. Note, these layers do not sum up to 100, unless each MS-NFI pixel within the LC-pixel is classified as forest. LC- pixels where the portion of forested pixels in MS-NFI data was less than 15% were removed from these layers to correspond with the enhanced LC-product. In addition, gapfilled pixels are provided as separate layers which allow users to either include or exclude these from computations. To construct a complete surface representation, the other land cover types (i.e. non-forested pixels) from the ESA LC-product are provided as a separate layer which can be summed with the percentage layers.