# Peer review of "An enhanced forest classification scheme for modeling vegetationclimate interactions based on national forest inventory data"

_Biogeosciences, 2017_

## Referee Comment (RC1) · Anonymous Referee #1 · 21 Aug 2017

Review of bg-2017-301: An enhanced forest classification scheme for modeling vegetation-climate interactions based on national forest inventory data

Summary

The authors present a species-based structural classification of forest types based on field and other data and apply it to a global land cover product. The more detailed forest classification considerable changes the forest cover in the global product.

This is an interesting project that has results that are potentially useful to a wider community, but there are several shortcomings that need to be addressed prior to being considered for publication. These are the main issues, with specific comments following:

1) The introduction lacks a literature review and does not present an appropriate context.

2) The methods are incomplete and confusing. The use of crown length and LAI seem extraneous and complicate the classification.

3) The significant difference between this study's classification and the global product is one of the major findings, yet it is not presented as such and the implications of it is not discussed

4) The discussion should discuss the implications of the findings rather than speculate about ecosystem modeling of structural attributes.

Specific comments and suggestions:

Abstract

Be specific that the key structural variables used are volume and lorey's height. It seems like another key finding here is how much the CCILC land cover classification changed. It appears that there was only ∼65% agreement between the CCI data and the NFI data.

Introduction

This introduction needs a context, and also a literature review.

It seems that linking forestry and earth system modeling is a desired context, but this is not developed. A context needs to presented and explained up front, showing where this classification fits in to improve earth system projections.

Actually, the better context here may be land cover classification, as that is what the project is, and there is a major difference between this study and the global product. Classification has been done at many scales with many methods, and here you merge

a couple of levels of data and products. Maybe this should be the focus of your literature review.

Also, there are little to no citations, but a lot of assertions. For example, management intensity affects forest structure, or climate modelers can't use forest information, or intervention and forest structure relationships are well known.

There are actually several dynamic vegetation models (no citations there either) that simulate succession and have height classes, and global data sets that include primary and secondary forest area and transitions.

page 2, line 9 LAI quantifies the areal exchange interface between vegetation and the atmosphere

Materials and Methods

The intro section 2.2 is too long and confusing. Make just a brief summary and rely on the figure for the overview, then go through the process in order. And section 2.3 is also part of the section 2.2 summary, which means it should be organized accordingly.

page 4, line 6 what type of clustering analysis?

page 4, lines 7-8 what classification scheme?

page 4, line 8 what are these maps? you mention later that they are land cover maps. But even later it sounds like they have more info than that. resolution? what are the data sources?

page 4, lines 8-11 this is confusing. are the MS-NFI maps made already, or do you make them? what data do you use for the extrapolation? what role do the MS-NFI maps play?

page 4, lines 14-15 what classified high resolution maps?

page 5, line 1 what about the decrease in the sum vs the increase in clusters determines the optimum? also, is this the number for each of the species groups?

page 5, line 31 what method did you use to assign the MS-NFI data to the classes? The lookup table just has single numbers for V and H (presumably the cluster centers?). the maps likely have a range of values.

page 6, lines 1-6 if you classify based on V and H, why do you originally cluster with CL and LAI? Especially since you correct the original clusters with V and H and MD? It seems like you have an extra, unnecessary step. This also appears to split up your subgroups in strange ways (see figure 2 for non-contiguous groups in VH space)

page 6, line12 you went from figure 1 to figure 4. please number your figures in the order of reference.

page 6, lines 8-19 what does this have to do with the structural classification? this seems to just be cover percent based on resampling the high-res forest cover.

page 7, lines 17-18 euclidean distances for 4 variables, not just V and H. in 2-d space the euclidean distance would give circular boundaries with contiguous groups. It isn't clear why CL and LAI were used.

page 8, line 4 reference your tables in numerical order

page 8, lines 1-19 there seems to be a very large difference between the CCI LC data and your data. This is a key finding and should be discussed with respect to which data may be more accurate, or whether this is simply a large level of land cover uncertainty.

Figures and Tables

Figure 3 What are the subgroups?

Table 4 is not referenced at all

---

## Referee Comment (RC2) · Anonymous Referee #2 · 10 Sep 2017

The manuscript deals with an important subject: improving the spatial characterisation of forest cover based on their structural properties by exploiting the rich information available in national forest inventories. Such work has the potential of improving land surface and earth system modelling predictions by providing a representation that is closer to reality than what is currently available. I therefore support the publication of this work, but I would require three main modifications that I believe would make the work stronger and more relevant.

First, the methodology should be clarified. I do not find all the steps to be crystal clear while reading it, even though if latter, by deduction, I end up understanding better the

reason for making them. To clarify things, I strongly suggest to separate the "material" from the "methods" part. By describing first the datasets used, I expect the flow of the methodology will naturally be improved. In the methods sections, avoid several levels of sub-headings: use only one. Also, at each step, try to start by specifically mentioning the objective of each step, e.g. "here we cluster the NFI plot data to get the mean structural properties (V, H, etc.) for our desired output map classes" and "here we use the spatializedy values of V and h from the MS-NFI maps along with the classifier of the previous step to make a map of our 12 forest types".

Second, the newer version of the CCI-Land Cover product should be used. The version the is currently used in this manuscript (v1.6) has been rendered obsolete by the current one (v2.0.7), as mentioned on the CCI website. There might be some classification errors that have been corrected in the newer version. Furthermore, for your map and analysis to stay relevant for longer, it would be highly valuable to be compatible with the more stable v2 product, which is much more likely to be adopted by modellers. Since the CCI_LC project has just finished, the latest version is probably the one the will be adopted more. This new CCI-LC consists of annual maps (instead of 5 year epochs), and thus a single year should be selected: probably the latest one (2015) or the one closest to the moment the MS-NFI maps represent.

Third, this paper could additionally serve explicitly as a regional validation of the CCI product. By this I suggest you actually frame this as a sub-objective of your work, i.e. not only provide an enhanced map, but also describe where and how much (and perhaps why) the original CCI wrong. This kind on feedback is useful for users in Fennoscadia, but also for the map producers to know how they can improve their global methods. In this vein, I would suggest to enhance figure 4 to add a second bar next to each bar in the histograms with the percentage of the original CCI map as well; and perhaps to add three columns of squared sub-figures on the right showing close-ups of (a) the original CCI maps, (b) the enhanced maps, and (c) the 12 class maps. This series of subplots could also be placed in another figure if necessary (but keeping

boxes on the figure 4 map to indicate their location. The objective is to show to the reader the impact of your reclassification on the spatial patterns at finer scale.

Here are some more punctual remarks on the text:

Page 3. Lines 10-11: While NFI data is available in many developed countries, these are not necessarily harmonised. Also, having 'gaps' in other countries could lead to further biasing global estimates of fluxes towards ecosystems of developed countries (i.e. the temperate northern hemisphere). What would you suggest to remediate this issue? (and perhaps discuss this in your discussions)

Page 4. Line 23: Why not Finland?

Page 5. Line 16: Not clear to me if these maps have spatializedy information of species type? or is it just V and h?

Page 6. Lines 32 until line 4 in [age 7: This description of the new ID codes is a bit too much of a 'metadata' that I would expect to find in the Supplementary Information (SI) section.

Page 7. Lines 28-31 and table 2: why not add some information of the spread of these values, such as the IQR or the range.

Page 9. Lines 1-3: One could argue that for Fennoscandia the forest are so monospecific that land cover classes are almost equal to species mapping, no?

Page 9. Line 20: How about VOD? see Liu. Y.Y.,et al (2013), Global Ecology and Biogeography and/or Konings, A. G., et al. (2017). Remote Sensing of Environment, doi.org/10.1016/j.rse.2017.06.037

Page 9. Line 12. Also, some very useful traits can be hard (or impossible) to map with remote sensing data (such as some traits regarding the roots for instance)

Page 10 line 31: Could you specify here to whom should the MS-NFI maps be requested?

SI. Page 1. Lines 5-15: not so clear to me, is dbh the only input to extract LAI?

SI. Page 1. Lines 16-19: is there any way to do some error propagation to provide some uncertainty (i.e +/- sigma) on the LAI values?

SI. Page 4. I wonder to what extent quite some of this background information should be in the main text (in the new "material" section)

---

## Referee Comment (RC3) · Anonymous Referee #3 · 11 Sep 2017

In the article "An enhanced forest classification scheme for modeling vegetation-climate interactions based on national forest inventory data", the authors present the method to upgrade the Land Cover (LC) product, a remote sensing derived LC map in the frame of European Space Climate Change Initiative (ESA CCI), with the national forest resources data for Norway, Sweden, and Finland. The work presented here has a great potential to bridge the gap between observation (remote sensing and field data) and modelling community, but it is not mature enough to be published in the presented form. A major argument for that is author's claim that their enhanced forest dataset "can improve climate predictions in intensively managed forested regions and is consistent with climate model routines that simulate the effects of land transitions through area-

based changes in vegetation cover". However, they do not provide any evidence for that assertion. There are several approaches to improve that:

1) Comaprison of their enhanced dataset and ESA-CCI-LC against other available LC maps for Fennoscandia.

2) It seems that authors confuse LC classes and Plant Functional Types (PFTs). Climate models do not use LC classes as their input, but PFTs and different models can employ different PFTs depending on the land surface processes implemented in the model. However, ESA-CCI-LC product is accompanied with the user tool that converts LC classes (via cross-walking table) into major (generic) PFTs, that later need to be adopted for particular climate model. In that process of conversion from LC classes to PFTs used in models information implemented in the enhanced ESA-CCI-LC classes might be lost. Therefore, comparison of major PFTs generated from original ESA-CCI-LC and enhanced ESA-CCI-LC is needed to see if the authors' efforts really can make an impact on climate model input data. And if not perhaps suggestions for improving cross-walking table (sometimes also called Look-Up Table (LUT), but it should not be confused with LUT that authors define in text) of the ESA-CCI-LC user tool can be made, so that enhancement of the ESA-CCI-LC data can be really seen by modelling community.

3) The only true test to estimate the impact of enhanced ESA-CCI-LC on climate models would be to perform regional climate simulations for Fennoscandia, and compare the results between two simulations with different Plant Functional Types (PFTs), derived from ESA-CCI LC and their enhanced ESA-CCI LC.

From all of the above, only point 3) might prove the main claim of this paper, but this is probably out of the scope for this paper and it should be a subject of another (modelling) study. However, for modellers to decide if is worth to conduct a modelling study it is necessary to know the difference between input PFTs datasets derived from enhancend and original ESA-CCL-LC. Therefore, in order to make valid contribution to

link modelling and forest observation community (as this seems to be more objective of the manuscript, rather than speculating about the improvement of climate predictions), points 1) and 2) are needed to confirm that effort. Furthermore, the method of classification described in the paper is not well documented. Method lacks in depth explanations and references are missing. For example, on several occasions, authors are quoting R routines with their cryptic abbreviations, but not providing any information or reference what is the basis for the algorithm used in that code.

Summing all of the above here are specific comments and suggestions to authors:

Abstract

Provide clear results of your study after major revision as suggested above by comparing generic PFTs derived from ESA-CCI-LC and enhanced ESA-CCI-LC, rather than speculation about an improvement of climate predictions.

Introduction

Research topic and objective should be clearly defined. However, here a description of LC classes in ESA-CCI-LC dataset and PFTs used in climate models is confusing. Page 2 lines 6-11: weak definition of PFTs is given with a couple of examples for PFT properties. More elaborate description or some reference is needed here. Page 2 lines 12-15: it looks like authors are using LC classes and PFTs interchangeably which makes confusion. Therefore it is not clear from the text that follows in the Introduction, if the objective is either to improve Look-Up Table (LUT) (by adding a new property in LUT that will indicate that forest is managed or not, or to improve some of the existing variables in LUT on the basis of field data) or to improve the ESA-CCI-LC. Make the introduction to the point, provide a clear overview of the research area and clearly state your objective.

Materials and method

Page 3 Line 22: Fig. 3 is referenced in the text before Fig.1. This is unusual, figure

numbers should increase monotonically otherwise, can confuse readers. Page 4 lines 13 – 20: ESA-CCI-LC data set is accompanied by the confidence level for each grid, i.e. an estimate of the accuracy that each grid is correctly classified. It might be worth to explore this field as well and if the forest is present on the ESA-CCI-LC grid with low confidence, perhaps it should be discarded as well. Page 6, line 10: The ESA-CCI-LC product does not contain PFTs, but LC classes. You can generate generic PFTs with user tool accompanying ESA-CCI-LC dataset. Page 6, line 12: Fig. 4 is referenced before Fig. 2. Page 6, line 10-18: Subclasses 61, 62, 71, 72 in ESA-CCI-LC dataset are regional and they are not available for the whole globe. It is not clear from the statement in the lines 15-16, if they are not available only in the data set that you were using or not available at all.

Results

Though it is not clearly stated so far, I got the impression that the main outcome of this study should be enhanced LC map, speculating that enhancement should also have an impact on climate simulations. However, the difference between the original and enhanced map is very short and confusing described on page 8 with Confusion matrix. There is no clear description how to interpret numbers in that table, and is quite confusing that the highest agreement between two datasets for LC class 70 is only 30.4%

Discussion

Page 8, line 21: not clear terminology. As far as I understand four key forests structural attributes have been used for adding forest LC classes to the ESA-CCI-LC. The discussion should be more on the significance of these results, and that is potential to improve PFTs maps used in climate and land surface modelling community. However, if the enhanced LC map would improve climate simulations or not remains speculations. This is for at least two reasons: 1) regional climate models operate on 1-10 km resolution, and 2) LC classes need to be converted into PFTs (by cross-walking procedure). The question is how different would these 2 PFT maps converted from enhanced and original LC maps appear after aggregation to coarser resolution and cross-walking. Therefore, I suggest authors rewrite the article and clearly explain what they have done or to perform an analysis as I have suggested above. The latter approach would certainly serve to link the climate modelling and forest observation community, as they seem to aspire in the manuscript.

―――――――――――――――――――

---

## Author Comment (AC1) · 2 Nov 2017

We would like to thank the three anonymous referees for their constructive suggestions and comments. Please find our responses to all points raised by the referees enclosed. Our responses are indented, marked with bullets, and the font is blue.

On behalf of the co-authors, Titta Majasalmi

Please also note the supplement to this comment: https://www.biogeosciences-discuss.net/bg-2017-301/bg-2017-301-AC1-supplement.pdf

[Figure]

**Supplement:**

The authors present a species-based structural classification of forest types based on field and other data and apply it to a global land cover product. The more detailed forest classification considerable changes the forest cover in the global product. This is an interesting project that has results that are potentially useful to a wider community, but there are several shortcomings that need to be addressed prior to being considered for publication. These are the main issues, with specific comments following: 1) The introduction lacks a literature review and does not present an appropriate context. 2) The methods are incomplete and confusing. The use of crown length and LAI seem extraneous and complicate the classification. 3) The significant difference between this study's classification and the global product is one of the major findings, yet it is not presented as such and the implications of it is not discussed 4) The discussion should discuss the implications of the findings rather than speculate about ecosystem modeling of structural attributes.

- 1) The introduction will be fully revised to explain the appropriate context, and we will
  include more citations to relevant literature. We agree that having more detailed introduction
  may help people who are not familiar with the subject to grasp the intended context.
- O 2) The methodology section will be revised by separating Materials from Methods, and by simplifying the description of methods. Both crown length (CL) and LAI are needed because they are the key structural input variables used in land models: CL is needed to calculate displacement height and canopy bottom height, and LAI quantifies the exchange surface area between land surface and atmosphere. Given the poor introduction and framing of our research objectives, it is understandable that Reviewer 1 is of the opinion that these variables seem extraneous. In our revision we will put forth a more convincing argument and rationale as to why we chose these variables, which represent important controls on photosynthesis, albedo, evapotranspiration, and other climate-relevant physical processes.
- 3) We fully agree that this is the most important and relevant finding of our study. The paper will be majorly revised to highlight the value of the enhanced LC product over the current product, which will be done by quantifying and mapping differences between the two products in the Results section together with an expanded discussion surrounding where and why the differences arise.
- 4) The reviewer raises a fair point that we have done a poor job of limiting the discussion only to claims supported by the results obtained in the paper. We will refrain from speculation in a revised manuscript that adds a new analysis that clearly demonstrates the added-value of our product by comparing a present day LAI map of Fennoscandia -- based on our enhanced LC product -- to LAI maps from several leading climate models which rely on the standard ESA LC -product. We believe such a demonstration would significantly strengthen the paper by helping the reader to understand the value of the enhanced LC-product and corresponding LUT.

Specific comments and suggestions:

**Abstract**

Be specific that the key structural variables used are volume and lorey's height. It seems like another key finding here is how much the CCI LC land cover classification changed. It appears that there was only \_65% agreement between the CCI data and the NFI data.

The key structural variables for which total within-cluster variance is minimized are H, V, CL, and max LAI (p4, r2-3). The structural variables for which cluster boundaries are defined and mapped in space are H and V (p5, r5-7). We will make this clear in a revised manuscript. We agree that differences between the LC-products should be presented as results in revised Results section and discussed in a revised Discussion section.

**Introduction**

This introduction needs a context, and also a literature review. It seems that linking forestry and earth system modeling is a desired context, but this is not developed. A context needs to presented and explained up front, showing where this classification fits in to improve earth system projections. Actually, the better context here may be land cover classification, as that is what the project is, and there is a major difference between this study and the global product. Classification has been done at many scales with many methods, and here you merge a couple of levels of data and products. Maybe this should be the focus of your literature review. Also, there are little to no citations, but a lot of assertions. For example, management intensity affects forest structure, or climate modelers can't use forest information, or intervention and forest structure relationships are well known. There are actually several dynamic vegetation models (no citations there either) that simulate succession and have height classes, and global data sets that include primary and secondary forest area and transitions.

• This suggested re-framing is good and we are happy to move forward with such a major revision as we fully agree that it will place the study in a better context. The reviewer is correct that the desired context is improvements to land cover classification for improved earth system modeling. As surface datasets in many climate and large-scale hydrological models are often based on land cover classifications like IGBP and UNLCCS (i.e, the ESA LC-product), the framing of our study should clearly explain the role of LC-classification and conversion table (i.e. from LC-classes to PFTs) on obtaining PFT-dependent initial parameters for model runs. The scientific quality of the revised Introduction section will be elevated with appropriate citations and will refrain from unsupported assertions.

**page 2, line 9 LAI quantifies the areal exchange interface between vegetation and the atmosphere**

• We will use the original definition: "hemisurface area of foliage per unit horizontal ground surface area" (Chen and Black, 1992) (doi:10.1111/j.1365-3040.1992.tb00992.x).

**Materials and Methods**

The intro section 2.2 is too long and confusing. Make just a brief summary and rely on the figure for the overview, then go through the process in order. And section 2.3 is also part of the section 2.2 summary, which means it should be organized accordingly.

• Materials and Methods (M&M) sections will be separated to clarify the data used following reviewer 2's suggestion. The redundant repetition will be removed and the text revised to follow the working order.

**page 4, line 6 what type of clustering analysis?**

• We used Euclidean distance based methods for defining the cluster medoids and cluster boundaries (as is described in M&M sections 2.2.1 (p4, r31) and 2.2.2 (p5, r8)).

**page 4, lines 7-8 what classification scheme?**

• We are referring here to our forest classification scheme. These lines will be removed to follow Reviewer 1's previous recommendations.

page 4, line 8 what are these maps? you mention later that they are land cover maps. But even later it sounds like they have more info than that. resolution? what are the data sources?

- The MS-NFI maps are forest resource maps processed by forest authorities. The processing of the maps is explained p4, r8-12: "MS-NFI maps extrapolate forest characteristics for areas between NFI field lots using a non-parametric k-Nearest Neighbor (kNN) estimation method (e.g. Tomppo et al., 2014). This extrapolation step is called "multi-source" because it employs data from different remote sensing systems (i.e. satellite and aerial platforms) and field plots. MS-NFI applies high resolution satellite images to separate forested areas from other LCcategories and digital terrain models to correct topographical distortions." (http://www.metla.fi/julkaisut/workingpapers/2014/mwp319.pdf).
- References, resolution, and other details on MS-NFI maps were provided in section 2.3.1. (p5.r16). MS-NFI maps contain only information related with forest resources, so they are not LC-products (i.e. data only for forested pixels). We could not find the place where MS-NFI maps are referred as land cover maps.
- Note, in a revised version of the manuscript the MS-NFI maps will be described in Materials section, which will help to clarify both data and processing.

**page 4, lines 8-11 this is confusing. are the MS-NFI maps made already, or do you make them? what data do you use for the extrapolation? what role do the MS-NFI maps play?**

We agree that the description of MS-NFI data is very scattered in this version of the manuscript – a revised manuscript will contain all information on MS-NFI maps in Materials section to help the reader. MS-NFI maps are products made by forest authorities and were used to apply our classification spatially. The processing in described in p4, r8-12 (See our previous answer).

**page 4, lines 14-15 what classified high resolution maps?**

• "Classified high resolution maps" refers to MS-NFI maps which have been classified using the enhanced forest classification scheme developed and presented in the paper (for clarity it could alternately be termed 'classified MS-NFI maps'). The unclear sentence will be removed to follow the recommendations.

**page 5, line 1 what about the decrease in the sum vs the increase in clusters determines the optimum? also, is this the number for each of the species groups?**

• The number of clusters was assessed based on the decrease in total within sum of squares as the number of clusters increased (i.e. 'elbow method') (p4-5, r32-1). We could revise this e.g. "The optimal number of clusters was assessed by plotting the curve between total within-cluster sum of squares (wcss) and the number of clusters (k), and observing around which k the relationship resamples a 'bend knee' (i.e. 'elbow method'). In each species group the bent knee was located between k=3 and k=5, and thus the optimal k was set to 4 (i.e. the number of structural subgroups was set to four). "

• We are a bit unsure as to what the reviewer means here. Species groups were formed based on largest share of total stem volume (m3/ha) (p3, r29-30), and within each species group (p4, r23-25) the structural subgroups were formed during the clustering analysis (p4-5, 32-3). This could be explained in the revised version as: "Then, using the predefined number of structural subgroups (k=4) within each species group (n=3), a K-medoids clustering analysis was used to define the cluster 'centroids' within V-H-CL-LAI -space to form LUT of the key forest structural attributes (n×k=12 classes). ".

page 5, line 31 what method did you use to assign the MS-NFI data to the classes? The lookup table just has single numbers for V and H (presumably the cluster centers?). the maps likely have a range of values.

- This is described in sections 2.2.1. and 2.2.2. The LUT was formed based on k-medoids clustering of the NFI data. Cluster memberships were determined using Mahalanobis Distance (MD). The LUT values are cluster medians (p4, r28-29). Application of the LUT requires maps of two variables (i.e. V and H), and cluster memberships-patterns (i.e. subgroup-memberships in VH-space) are visualized in Figure 2.
- Methods section will be fully be revised to improve readability. We agree that we should clarify that we were actually using two LUTs per species: one for defining the cluster median values (to form the LUT of forest key structural attributes) and another one for classifying the MS-NFI data into predefined classes (in gridded VH-space i.e. Figure 2.).

page 6, lines 1-6 if you classify based on V and H, why do you originally cluster with CL and LAI? Especially since you correct the original clusters with V and H and MD?

- The LUT values represent medoids of the four-dimensional (i.e. V-H-CL-LAI) clusters of the twelve forest classes (3×species group × 4×subgroup). Related LC-product is needed to apply the LUT to obtain Fennoscandic maps of V, H, CL and LAI. The LC-map to apply the LUT was prepared using MS-NFI maps, and MS-NFI maps were classified based on V and H to simplify the classification task (p5, r6-7).
- The clusters are not corrected using MD, but it is used to determine cluster-membership (i.e. into which group an observation with certain V and H combination belongs to based on smallest median MD (i.e. for each subgroup a gridded representation median MD values was calculated, and the Fig. 2. was formed by assigning each grid cell location for the subgroup which had the smallest median MD) (p5, r7-13)

It seems like you have an extra, unnecessary step. This also appears to split up your subgroups in strange ways (see figure 2 for non-contiguous groups in VH space)

 We agree that two-dimensional visualizations of the four-dimensional clusters look odd. Fig. 2. was only used to assign MS-NFI pixels with different species and V-H-combinations into different subgroups. No unnecessary steps are present. Our approach essentially links the important structural parameters directly to the land cover classification.

page 6, line12 you went from figure 1 to figure 4. please number your figures in the order of reference.

• The reference to Fig. 4. may be removed.

page 6, lines 8-19 what does this have to do with the structural classification? This seems to just be cover percent based on resampling the high-res forest cover.

• This section introduces the ESA LC-product and its features. The ESA LC-product will be introduced in Materials section in the revised version of the manuscript.

page 7, lines 17-18 euclidean distances for 4 variables, not just V and H. in 2-d space the euclidean distance would give circular boundaries with contiguous groups. It isn't clear why CL and LAI were used.

- As explained earlier, numerical weather prediction, climate, and hydrological models require important structural inputs like H, CL (or Ztop and Zbottom), and LAI to resolve surface fluxes --As those variables are not independent from each other, the clustering had to be done in multiple dimensions to compile the LUT of these key structural attributes.
- The cluster median values of V, H, CL and LAI were required to form the LUT of forest key structural attributes, and gridded representation of the VH-space was only used to assign the MS-NFI data into predefined classes. These two steps are not independent, because for MD calculation the cluster mean values were obtained based on the 4d (i.e. V, H, CL and LAI) and not in 2d (i.e. V and H) clusters, and thus the boundaries are not circular. We agree this must be explained better in the revised manuscript.

page 8, line 4 reference your tables in numerical order

• Table 4 may be removed (or moved to SI) and remaining tables renamed.

page 8, lines 1-19 there seems to be a very large difference between the CCI LC data and your data. This is a key finding and should be discussed with respect to which data may be more accurate, or whether this is simply a large level of land cover uncertainty.

• We agree that the difference between ESA LC-product and our enhanced product should be analyzed and discussed more. We will add a difference map between ESA LC-product and our product to highlight areas where the largest differences in forest cover occur. In addition, a discussion on observed differences and their possible causes will be added. This might be interesting for those who develop LC-products.

**Figures and Tables**

**Figure 3 What are the subgroups?**

• The groups are the same as in Figure 2 (i.e. species group + subgroup number). This will be added to figure caption.

**Table 4 is not referenced at all**

• We will remove it.

**Anonymous Referee #2**

The manuscript deals with an important subject: improving the spatial characterisation of forest cover based on their structural properties by exploiting the rich information available in national forest inventories. Such work has the potential of improving land surface and earth system modelling predictions by providing a representation that is closer to reality than what is currently available. I therefore support the publication of this work, but I would require three main modifications that I believe would make the work stronger and more relevant. First, the methodology should be clarified. I do not find all the steps to be crystal clear while reading it, even though if latter, by deduction, I end up understanding better the reason for making them. To clarify things, I strongly suggest to separate the "material" from the "methods" part. By describing first the datasets used, I expect the flow of the methodology will naturally be improved. In the methods sections, avoid several levels of subheadings: use only one. Also, at each step, try to start by specifically mentioning the objective of each step, e.g. "here we cluster the NFI plot data to get the mean structural properties (V, H, etc.) for our desired output map classes" and "here we use the spatializedy values of V and h from the MS-NFI maps along with the classifier of the previous step to make a map of our 12 forest types". Second, the newer version of the CCI-Land Cover product should be used. The version the is currently used in this manuscript (v1.6) has been rendered obsolete by the current one (v2.0.7), as mentioned on the CCI website. There might be some classification errors that have been corrected in the newer version. Furthermore, for your map and analysis to stay relevant for longer, it would be highly valuable to be compatible with the more stable v2 product, which is much more likely to be adopted by modellers. Since the CCI\_LC project has just finished, the latest version is probably the one the will be adopted more. This new CCI-LC consists of annual maps (instead of 5 year epochs), and thus a single year should be selected: probably the latest one (2015) or the one closest to the moment the MS-NFI maps represent. Third, this paper could additionally serve explicitly as a regional validation of the CCI product. By this I suggest you actually frame this as a sub-objective of your work, i.e. not only provide an enhanced map, but also describe where and how much (and perhaps why) the original CCI wrong. This kind on feedback is useful for users in Fennoscadia, but also for the map producers to know how they can improve their global methods. In this vein, I would suggest to enhance figure 4 to add a second bar next to each bar in the histograms with the percentage of the original CCI map as well; and perhaps to add three columns of squared sub-figures on the right showing close-ups of (a) the original CCI maps, (b) the enhanced maps, and (c) the 12 class maps. This series of subplots could also be placed in another figure if necessary (but keeping boxes on the figure 4 map to indicate their location. The objective is to show to the reader the impact of your reclassification on the spatial patterns at finer scale.

- We agree that we have failed to see our paper as an opportunity to validate (regionally) the forest extent information of the ESA LC-product. Reviewer 2's idea to investigate the differences between the two LC-products is good, and will be outlined as one of the research objectives. We will prepare a difference map between ESA LC-product forest classes and enhanced LC-product forest classes. The difference map will show the areas where the largest differences occur between MS-NFI forest and that defined by ESA LC, and relevant discussion on why the differences occur will be added.
- Reviewer 2 makes a good point by recommending the use of a more stable v.2.0.7. product. In a revised manuscript we will use the newer v.2.0.7. ESA LC-product instead of the v.1.6. LC-product which was used in the first version.

- We agree that separating Materials and Methods (M&M) sections will clarify the storyline.
- The M&M sections may be formatted to have only one-level of subheadings, and the objective of each step can be mentioned at each step to help the reader.
- We will include histogram-bars of the original ESA LC-product to Fig. 4., so that the class frequencies can be visually compared. Class changes (e.g. from forest to a cropland) can be seen from the confusion matrix. We believe that analysis of forest cover information will be useful for both the product users and map developers.
- However, we think that adding subfigures to figure 4 is not practical, because we would also need to add legends (e.g. ESA LC-product legend has 27 classes, the enhanced LC-product has 71 classes (i.e. containing special coding which indicates within-pixel forest cover information, and if pixel is forest based on MS-NFI data 'true forest' or gapfilled), or 35 classes if information on forest cover and gapfilling are excluded). The spatial distribution of 'true forest' pixels is visualized already in Fig. 3).

**Here are some more punctual remarks on the text:**

Page 3. Lines 10-11: While NFI data is available in many developed countries, these are not necessarily harmonised. Also, having 'gaps' in other countries could lead to further biasing global estimates of fluxes towards ecosystems of developed countries (i.e. the temperate northern hemisphere). What would you suggest to remediate this issue? (and perhaps discuss this in your discussions)

It is true that data and methods used in NFIs between different countries are not necessarily harmonized due to different tree species and funding available for conducting field inventories. However, nowadays most countries are conducting NFIs to quantify the extent and amount of forest resources with standardized reporting e.g. for Food and Agriculture Organization of the United Nations for compiling global Forest Resources Assessments – thus, there exists ways to standardize the data. We raise this issue in the Introduction and Discussion sections, but indepth discussion on data standardization is outside the scope of this paper.

**Page 4. Line 23: Why not Finland?**

• We employed NFI plot data only from Norway and Sweden, but MS-NFI maps for all countries. As we state in the Methods section (p.3, r22-23), there is little additional value by including Finnish plot-level NFI data given that forest structure and composition in Finnish and Swedish forests is similar.

**Page 5. Line 16: Not clear to me if these maps have spatializedy information of species type? or is it just V and h?**

We will separate the Materials and Methods sections and revise the description of the MS-NFI data to clarify the data and processing. MS-NFI maps have spatialized information on forest variables such as Lorey's height (i.e. basal area weighted mean tree height) ('H', m), forest stand age (years), and stem volume ('V', m3/ha) by species.

Page 6. Lines 32 until line 4 in [age 7: This description of the new ID codes is a bit too much of a 'metadata' that I would expect to find in the Supplementary Information (SI) section.

• We agree that this information would be better-placed in the SI.

Page 7. Lines 28-31 and table 2: why not add some information of the spread of these values, such as the IQR or the range.

• We will add IQR -values to Table 2.

Page 9. Lines 1-3: One could argue that for Fennoscandia the forest are so monospecific that land cover classes are almost equal to species mapping, no?

• Unfortunately, the issue is that pure stands of single species are relatively scarce. Forests are mainly mixtures of different species.

Page 9. Line 20: How about VOD? see Liu. Y.Y., et al (2013), Global Ecology and Biogeography and/or Konings, A. G., et al. (2017). Remote Sensing of Environment, doi.org/10.1016/j.rse.2017.06.037

• We could add discussion surrounding the potential applicability of SAR data in this context. However, as the spatial resolution of the SAR data is low (many km2), such data may not be optimal in areas which are fragmented into 1-2 ha units by forest management.

Page 9. Line 12. Also, some very useful traits can be hard (or impossible) to map with remote sensing data (such as some traits regarding the roots for instance)

 $\circ$  Good point – this will be added to the text.

**Page 10 line 31: Could you specify here to whom should the MS-NFI maps be requested?**

• Thank you for pointing out the missing information. This will be included in a revised version of the manuscript.

**SI. Page 1. Lines 5-15: not so clear to me, is dbh the only input to extract LAI?**

- No, allometric tree-wise foliage mass models were first used to estimate total foliage mass of trees within a sample plot, and LAI was calculated by dividing plot total foliage mass with plot area and multiplying with Specific Leaf Area (SLA) value (i.e. conversion from mass to area)(SI p1, r15-16). We will start the section by clarifying this.
- In addition, the sentence (SI p1, r6-8) mentioning the inputs of allometric models can be revised as: "Marklund's tree dbh- and h -dependent models were used to calculate foliage mass of pine and spruce trees (T-18 and G-16, respectively), whereas for small trees the foliage mass was calculated using Marklund's dbh-dependent models (T-17 and G-15)."

**SI. Page 1. Lines 16-19: is there any way to do some error propagation to provide some uncertainty (i.e +/- sigma) on the LAI values?**

Unfortunately error propagation of LAI estimates is not possible, because the 'true' LAI is not known (for details see Majasalmi et. al., 2013) (DOI: 10.1016/j.foreco.2012.12.017). We will mention this is text.

**SI. Page 4. I wonder to what extent quite some of this background information should be in the main text (in the new "material" section)**

• We think that proper description of the MS-NFI data in Materials section will help readers to understand what the MS-NFI data is like. We will also discuss on where and why the ESA

LC-product and our enhanced LC-product forest extents differ the most, and thus some parts could be moved into main text.

**Anonymous Referee #3**

In the article "An enhanced forest classification scheme for modeling vegetation-climate interactions based on national forest inventory data", the authors present the method to upgrade the Land Cover (LC) product, a remote sensing derived LC map in the frame of European Space Climate Change Initiative (ESA CCI), with the national forest resources data for Norway, Sweden, and Finland. The work presented here has a great potential to bridge the gap between observation (remote sensing and field data) and modelling community, but it is not mature enough to be published in the presented form. A major argument for that is author's claim that their enhanced forest dataset "can improve climate predictions in intensively managed forested regions and is consistent with climate model routines that simulate the effects of land transitions through area based changes in vegetation cover". However, they do not provide any evidence for that assertion. There are several approaches to improve that: 1) Comaprison of their enhanced dataset and ESA-CCI-LC against other available LC maps for Fennoscandia. 2) It seems that authors confuse LC classes and Plant Functional Types (PFTs). Climate models do not use LC classes as their input, but PFTs and different models can employ different PFTs depending on the land surface processes implemented in the model. However, ESA-CCI-LC product is accompanied with the user tool that converts LC classes (via cross-walking table) into major (generic) PFTs, that later need to be adopted for particular climate model. In that process of conversion from LC classes to PFTs used in models information implemented in the enhanced ESA-CCI-LC classes might be lost. Therefore, comparison of major PFTs generated from original ESA-CCILC and enhanced ESA-CCI-LC is needed to see if the authors' efforts really can make an impact on climate model input data. And if not perhaps suggestions for improving cross-walking table (sometimes also called Look-Up Table (LUT), but it should not be confused with LUT that authors define in text) of the ESA-CCI-LC user tool can be made, so that enhancement of the ESA-CCI-LC data can be really seen by modelling community. 3) The only true test to estimate the impact of enhanced ESA-CCI-LC on climate models would be to perform regional climate simulations for Fennoscandia, and compare the results between two simulations with different Plant Functional Types (PFTs), derived from ESA-CCI LC and their enhanced ESA-CCI LC. From all of the above, only point 3) might prove the main claim of this paper, but this is probably out of the scope for this paper and it should be a subject of another (modelling) study. However, for modellers to decide if is worth to conduct a modelling study it is necessary to know the difference between input PFTs datasets derived from enhancend and original ESA-CCL-LC. Therefore, in order to make valid contribution to link modelling and forest observation community (as this seems to be more objective of the manuscript, rather than speculating about the improvement of climate predictions), points 1) and 2) are needed to confirm that effort. Furthermore, the method of classification described in the paper is not well documented.

As the only difference between our product and the ESA LC-product is the definition and description of a forest (i.e. other LC-classes were brought from ESA LC-product), there is no need for doing comprehensive LC-validation. Note, as we will use the newest ESA LC-product (v.2.0.7) following Reviewer 2's suggestion – and that product has been validated against GlobCover 2009 (ESA LC 2015 product manual (p.39)) – we think it is enough to mention this in text and cite the source file. However, we will make a difference map between ESA LC-product (v.2.0.7.) forest classes and enhanced LC-product forest classes to point out areas where the largest differences occur between MS-NFI forest and that defined by ESA LC.

- As only the definition of forest differs between the two LC-products, we cannot really suggest any improvements for the cross-walking-table (i.e. our product does not need any and thus there is no need for developing user-tools).
- 2) The introduction section will be revised to explain how LC-products and cross-walking tables are used to convert between LC-classes and PFTs. We agree that this should have been explained more carefully at the onset for the benefit of the modeling audience, and our revision will make clear how our LUT differs to PFT LUTs used in cross-walking exercise by modelers (i.e., Poulter et al. 2015 https://doi.org/10.5194/gmd-8-2315-2015).
- We agree with Reviewer 3 that, while addressing concern number 3) is outside the scope of this paper, a more robust effort to address concerns 1) and 2) would be enough to allay concerns that the value of our enhanced LC-product has not been properly demonstrated or critically evaluated. To this extent, we will add a new analysis that compares the spatial distribution of max LAI in Fennoscandic forests based on our product and the related LUT compared to that obtained using standard ESA LC-product and PFT-dependent max LAI values of three reputable land models which recently participated in the development of the harmonized ESA LC-PFT cross-walking method presented in Poulter et al., (2015).
- As mentioned above, new analysis to address main concern number 2) will be undertaken to build credibility and elevate the overall value of our research contribution. In other words, a revised manuscript will present a more convincing argument that our enhanced LC-product with its related LUT of LC-dependent structural attributes can better account for local variations in important structural parameters compared to existing schemes based on the standard ESA LC product. From this, our initial claim that the novel approach "can lead to better predictions of surface fluxes" can be inferred more readily.

Method lacks in depth explanations and references are missing. For example, on several occasions, authors are quoting R routines with their cryptic abbreviations, but not providing any information or reference what is the basis for the algorithm used in that code.

• We can revise description of the methods and add citations to used algorithms/methods.

Summing all of the above here are specific comments and suggestions to authors:

**Abstract**

Provide clear results of your study after major revision as suggested above by comparing generic PFTs derived from ESA-CCI-LC and enhanced ESA-CCI-LC, rather than speculation about an improvement of climate predictions.

• The abstract will be revised to state the results.

**Introduction**

Research topic and objective should be clearly defined. However, here a description of LC classes in ESA-CCI-LC dataset and PFTs used in climate models is confusing.

• We will reformate the research questions, and fully revise Introduction. We will clarify how LC-products and cross-walking tables are required for obtaining PFT-distributions employed by large-scale climate/hydrological models.

Page 2 lines 6-11: weak definition of PFTs is given with a couple of examples for PFT properties. More elaborate description or some reference is needed here.

• We will add the definition of PFTs used in this study: "broad groupings of plant species that share similar characteristics (e.g. growth form) and roles (e.g. photosynthetic pathway) in ecosystem function" (Wullschleger et al., 2014) (https://doi.org/10.1093/aob/mcu077).

Page 2 lines 12-15: it looks like authors are using LC classes and PFTs interchangeably which makes confusion. Therefore it is not clear from the text that follows in the Introduction, if the objective is either to improve Look-Up Table (LUT) (by adding a new property in LUT that will indicate that forest is managed or not, or to improve some of the existing variables in LUT on the basis of field data) or to improve the ESA-CCI-LC. Make the introduction to the point, provide a clear overview of the research area and clearly state your objective.

• The entire introduction will be revised and the research objectives stated more clearly. We will clarify terminology in the revised version of the paper.

Materials and method Page 3 Line 22: Fig. 3 is referenced in the text before Fig.1. This is unusual, figure numbers should increase monotonically otherwise, can confuse readers.

• We will remove reference to Fig. 3.

Page 4 lines 13 - 20: ESA-CCI-LC data set is accompanied by the confidence level for each grid, i.e. an estimate of the accuracy that each grid is correctly classified. It might be worth to explore this field as well and if the forest is present on the ESA-CCI-LC grid with low confidence, perhaps it should be discarded as well.

 Evaluating the accuracy of the current ESA LC product is beyond the scope of this paper, as our objective was simply to augment the representation of forest cover only using local MS-NFI information. We will in a revised manuscript, however, evaluate the accuracy of the forest cover of the current ESA LC product by comparing it with the MS-NFI data compiled by regional mapping authorities.

Page 6, line 10: The ESA-CCI-LC product does not contain PFTs, but LC classes. You can generate generic PFTs with user tool accompanying ESA-CCI-LC dataset.

• The definition of PFT will be amended throughout the manuscript to align with that used by the modeling community and Reviewer 3.

Page 6, line 12: Fig. 4 is referenced before Fig. 2.

• Reference to Fig. 4 will be removed.

Page 6, line 10-18: Subclasses 61, 62, 71, 72 in ESA-CCI-LC dataset are regional and they are not available for the whole globe. It is not clear from the statement in the lines 15-16, if they are not available only in the data set that you were using or not available at all.

• This information will be added to text.

Results

Though it is not clearly stated so far, I got the impression that the main outcome of this study should be enhanced LC map, speculating that enhancement should also have an impact on climate simulations. However, the difference between the original and enhanced map is very short and confusing described on page 8 with Confusion matrix. There is no clear description how to interpret numbers in that table, and is quite confusing that the highest agreement between two datasets for LC class 70 is only 30.4%.

- Yes, the main outcome of this study is the enhanced LC-product and the related LUT of the key structural attributes (and also the percentage layers which allow more flexibility in PFT cross-walking or re-sampling between different resolutions).
- As previously mentioned, we will add a new figure to point out areas where the forest extent information differs between the enhanced LC-product and current ESA LC-product, and to discuss the reasons why these differences occur. In addition, we will demonstrate how our enhanced LC-product and the related LUT may be used to map local variations in important structural attributes, such as max LAI, contrasting these to those produced by three major land models when run in dynamic vegetation mode.
- Confusion matrix is a standard way of comparing two LC classifications. We will add more details on how to interpret the confusion matrix to help the readers.

**Discussion**

Page 8, line 21: not clear terminology. As far as I understand four key forests structural attributes have been used for adding forest LC classes to the ESA-CCI-LC.

• This unclear sentence will be removed and the discussion rephrased. We prepared our own LC-product (and the related LUT) which was supplemented with LC data from ESA LC-product.

The discussion should be more on the significance of these results, and that is potential to improve PFTs maps used in climate and land surface modelling community. However, if the enhanced LC map would improve climate simulations or not remains speculations.

• Discussion will be rephrased to fit with the new Results sections suggested by the reviewers (and thus the amount of speculation will be reduced).

This is for at least two reasons: 1) regional climate models operate on 1-10 km resolution, and 2) LC classes need to be converted into PFTs (by cross-walking procedure). The question is how different would these 2 PFT maps converted from enhanced and original LC maps appear after aggregation to coarser resolution and cross-walking. Therefore,I suggest authors rewrite the article and clearly explain what they have done or to perform an analysis as I have suggested above. The latter approach would certainly serve to link the climate modelling and forest observation community, as they seem to aspire in the manuscript.

As our enhanced LC-product has an attached LUT of the key forest structural attributes, no cross-walking procedure is needed, i.e., our LUT may be applied to forest classified pixels directly. Cross-walking is only needed to convert the ESA LC-product to a PFT-map. The influence of data aggregation to coarser resolution on raster values will be analyzed and discussed in the revised manuscript. We agree that that our paper needs substantial revision, and that the suggested corrections and analysis steps will increase the impact of our paper.

---

## Author Response (AR1)

November 29, 2017

Titta Majasalmi

1431 Ås, Norway

titta.majasalmi@nibio.no

JOURNAL OF BIOGEOSCIENCES

Editorial office

Dear Associate Editor, Kirsten Thonicke,

Please find enclosed our revised manuscript "An enhanced forest classification scheme for modeling vegetation-climate interactions based on national forest inventory data", by Titta Majasalmi, Stephanie Eisner, Rasmus Astrup, Jonas Fridman and Ryan M. Bright. We are grateful for your comments and have invested substantial attention and effort to ensuring that yours -- along with and other reviewer comments -- have been carefully addressed. You will find that all sections in the manuscript were revised following the reviewers suggestions, and we believe the quality of the paper is greatly improved.

On behalf of all authors,

sincerely,

Titta Majasalmi

Corresponding author

Comments to the Author:

Thank you for submitting your carefully formulated author responses! I think it is very important to address the 4 major points of reviewer 1 in the revised manuscript as you rightly have announced.

- The Introduction and Materials and Methods (M&M) were fully revised. Introduction presents the appropriate context, and in M&M data and methods were separated. The role of crown length (CL) and Leaf Area Index (LAI) in modeling surface fluxes is explained in Introduction (p.2: r. 5-10), and the fact that the forest variables are not independent from each other is explained in M&M (p.5: r. 24-26). The differences between our Land Cover (LC) -product and the ESA LC-product are now presented in results section (**Fig. 4.**, **Fig. 5.** and **Table 4.**, and discussed on Discussion section (p.12-13: r. 20-2). In addition, discussion was revised to correspond with the new result section (e.g. p.13-14: r. 18-2).

It is also important to use the latest version of CCI_LC product as you agreed in your response to reviewer 2. In my view it is ok not to add subfigures in Fig. 4 (response to reviewer 2).

- The newest ESA LC-product (v.2.0.7) was used.

I also agree with you that it is important to provide the information where the MS-NFI maps can be requested. Would it be possible to provide them open access, given that you get a data DOI before?

- The missing information for requesting MS-NFI data for Norway was added (p.14: r. 22-23). Our products (i.e. the enhanced LC-product and the respective percentage layers will have open access and doi. Requesting doi means that no further changes of the files are possible – so we must wait until our paper is accepted before requesting the doi).

I have a few comments to make:

Response to Reviewer 1:

1. Please also double-check that the revised manuscript text adequately explains this point:

R1: page 6, lines 1-6 if you classify based on V and H, why do you originally cluster with CL and LAI? Especially since you correct the original clusters with V and H and MD?

Your response: "The LUT values represent medoids of the four-dimensional (i.e. V-H-CL-LAI) clusters of the twelve forest classes (3×species group × 4×subgroup). Related LC-product is needed to apply the LUT to obtain Fennoscandic maps of V, H, CL and LAI. The LC-map to apply the LUT was prepared using MS-NFI maps, and MS-NFI maps were classified based on V and H to simplify the classification task (p5, r6-7).

o The clusters are not corrected using MD, but it is used to determine cluster-membership (i.e. into which group an observation with certain V and H combination belongs to based on smallest median MD (i.e. for each subgroup a gridded representation median MD values was calculated, and the Fig. 2. was formed by assigning each grid cell location for the subgroup which had the smallest median MD) (p5, r7-13)"

- The M&M section was revised (p.5-6: r. 20-22). First paragraph introduces the variables and why they were chosen for clustering. Second paragraph explains how number of clusters was determined and how the clustering was done. Third paragraph describes how MD was applied to solve cluster memberships.

- The reason for incorporating all four variables into clustering analysis is explained in line (p.5: r. 24-26) (i.e. the variables are not independent from each other). V and H were only used to define cluster memberships in order to apply the classification for MS-NFI map data as pointed out in p.5: r. 26-27. MD was only used to assign each V-H-combination into one of the four-dimensional clusters and is described in p.6: r.13-22.

2. It will be important to provide information on the algorithms used in addition to the R routines, cf. Reviewer #3 comment:

R3: Method lacks in depth explanations and references are missing. For example, on several occasions, authors are quoting R routines with their cryptic abbreviations, but not providing any information or reference what is the basis for the algorithm used in that code.

Your response: "We can revise description of the methods and add citations to used algorithms/methods."

- We have added references and clarifications.
- For example, to determine the optimal number of clusters we enhanced the method description (p.5-6: r. 29-3) and reference for R package was added (p.6: r. 2).
- The description of the clustering was also modified and references were added to both for method (p.6: r. 7-8) and R package (p6: r. 11). The clustering method and application is explained in (p.6: r. 3-11).
- A reference was added to describe MD (p.6: r. 15). The definition of cluster boundaries is explained in p.6: r. 17-22.
- We added references to 'rgdal' and 'raster' packages (p.7: r. 19) (i.e. they only contain basic tools for reading, processing and analyzing geospatial data in R).
- For confusion matrix calculation, we added reference to 'caret' package (p.8: r. 3) and revised the section to explain how to interpret the confusion matrix (p.7-8: r. 31-3) (the information was also added to **Table 4.** Figure caption for reader's convenience).

3. in your revision of the introduction as well as Material and Methods, please make sure that the land-cover classes are clearly separated from PFTs in their definition.

- This was clarified throughout the manuscript.

4. Please make sure this point is adequately addressed (i.e. clearly described and discussed) in the revised manuscript.

R3: "This is for at least two reasons: 1) regional climate models operate on 1-10 km resolution, and 2) LC classes need to be converted into PFTs (by cross-walking procedure). The question is how different would these 2 PFT maps converted from enhanced and original LC maps appear after aggregation to coarser resolution and cross-walking. Therefore,I suggest authors rewrite the article and clearly explain what they have done or to perform an analysis as I have suggested above. The latter approach would certainly serve to link the climate modelling and forest observation community, as they seem to aspire in the manuscript."

Your response: "As our enhanced LC-product has an attached LUT of the key forest structural attributes, no cross-walking procedure is needed, i.e., our LUT may be applied to forest classified pixels directly. Cross-walking is only needed to convert the ESA LC-product to a PFT-map. The influence of data aggregation to coarser resolution on raster values will be analyzed and discussed in

the revised manuscript. We agree that that our paper needs substantial revision, and that the suggested corrections and analysis steps will increase the impact of our paper."

- The resolution difference between different LC-products is described/brought up in Introduction (p.3: r. 11-12), in M&M (p.7: r. 22-25), and included in Discussion (p.12: r. 26-32).
- The representation of LC classes in land models and LC-class to PFT conversion (i.e. cross-walking) is introduced in Introduction (p.2: r. 10-17).
- To follow the reviewers idea we decided to demonstrate the potential of our enhanced LC-product to improve the description of maximum growing season LAI ($LAI_{max}$) of managed forests in Fennoscandia – we compared our $LAI_{max}$ map with reference $LAI_{max}$ maps created using the ESA LC-product (and related cross-walking table) and PFT-dependent $LAI_{max}$ values used in three leading land models. The new section was added to M&M (p.8: r. 5-26), to results (p.10: r. 19-27), and to Discussion sections (p.13: r. 21-29).
- As both LC-products have the same spatial resolution, data aggregation to lower resolution is not needed to show that the two LC-products and different PFT-dependent $LAI_{max}$ values results in differently looking maps of $LAI_{max}$ (i.e. our $LAI_{max}$ map shows clear spatial patterns, and $LAI_{max}$ variation with latitude, whereas the reference $LAI_{max}$ maps remain constant).
- We are confident that this section now sufficiently answers R3's question of: "how different would 2 PFT maps converted from enhanced and original LC maps appear after cross-walking?"